# Nanofertilizers: Types, Delivery and Advantages in Agricultural Sustainability

Anurag Yadav [1,*], Kusum Yadav [2] and Kamel A. Abd-Elsalam [3]

1 Department of Microbiology, College of Basic Science and Humanities, Sardarkrushinagar Dantiwada Agricultural University, Sardarkrushinagar, District Banaskantha, Gujarat 385506, India
2 Department of Biochemistry, University of Lucknow, Lucknow 226007, India
3 Plant Pathology Research Institute, Agricultural Research Center, Giza 12619, Egypt
* Correspondence: anuragyadav123@sdau.edu.in or anuragyadav123@gmail.com

**Abstract:** In an alarming tale of agricultural excess, the relentless overuse of chemical fertilizers in modern farming methods have wreaked havoc on the once-fertile soil, mercilessly depleting its vital nutrients while inflicting irreparable harm on the delicate balance of the surrounding ecosystem. The excessive use of such fertilizers leaves residue on agricultural products, pollutes the environment, upsets agrarian ecosystems, and lowers soil quality. Furthermore, a significant proportion of the nutrient content, including nitrogen, phosphorus, and potassium, is lost from the soil (50–70%) before being utilized. Nanofertilizers, on the other hand, use nanoparticles to control the release of nutrients, making them more efficient and cost-effective than traditional fertilizers. Nanofertilizers comprise one or more plant nutrients within nanoparticles where at least 50% of the particles are smaller than 100 nanometers. Carbon nanotubes, graphene, and quantum dots are some examples of the types of nanomaterials used in the production of nanofertilizers. Nanofertilizers are a new generation of fertilizers that utilize advanced nanotechnology to provide an efficient and sustainable method of fertilizing crops. They are designed to deliver plant nutrients in a controlled manner, ensuring that the nutrients are gradually released over an extended period, thus providing a steady supply of essential elements to the plants. The controlled-release system is more efficient than traditional fertilizers, as it reduces the need for frequent application and the amount of fertilizer. These nanomaterials have a high surface area-to-volume ratio, making them ideal for holding and releasing nutrients. Naturally occurring nanoparticles are found in various sources, including volcanic ash, ocean, and biological matter such as viruses and dust. However, regarding large-scale production, relying solely on naturally occurring nanoparticles may not be sufficient or practical. In agriculture, nanotechnology has been primarily used to increase crop production while minimizing losses and activating plant defense mechanisms against pests, insects, and other environmental challenges. Furthermore, nanofertilizers can reduce runoff and nutrient leaching into the environment, improving environmental sustainability. They can also improve fertilizer use efficiency, leading to higher crop yields and reducing the overall cost of fertilizer application. Nanofertilizers are especially beneficial in areas where traditional fertilizers are inefficient or ineffective. Nanofertilizers can provide a more efficient and cost-effective way to fertilize crops while reducing the environmental impact of fertilizer application. They are the product of promising new technology that can help to meet the increasing demand for food and improve agricultural sustainability. Currently, nanofertilizers face limitations, including higher costs of production and potential environmental and safety concerns due to the use of nanomaterials, while further research is needed to fully understand their long-term effects on soil health, crop growth, and the environment.

**Keywords:** nanofertilizers; controlled release; delivery system; types

## 1. Introduction

As a cornerstone of sustaining the ever-growing global population and driving the thriving economy, agriculture assumes a vital role. In this pursuit, the indispensable use of fertilizers has emerged as an essential practice for augmenting crop yields and preserving soil fertility. Conventional fertilizers, such as urea, nitrogen, phosphorous, potassium, monoammonium phosphate, and diammonium phosphate, are widely utilized to supplement essential nutrients in the soil. However, conventional fertilizers suffer from low nutrient utilization efficiency due to leaching, leading to substantial economic losses and decreased soil fertility. The leaching of these nutrients from the soil has resulted in a significant decrease in soil fertility. This is primarily due to the relatively low nutrient utilization efficiency of conventional fertilizers, which is around 30–35% for nitrogen, 18–20% for phosphorus, and 35–40% for potassium [1] (Figure 1). The scientific community is already working on developing slow-release chemical fertilizers; for example, combining hydroxyapatite with urea has allowed researchers to develop slow-release fertilizers that gradually release plant nutrients [2]. Moreover, the environmental impact caused by releasing excess nutrients has necessitated the development of more efficient and eco-friendly fertilizers.

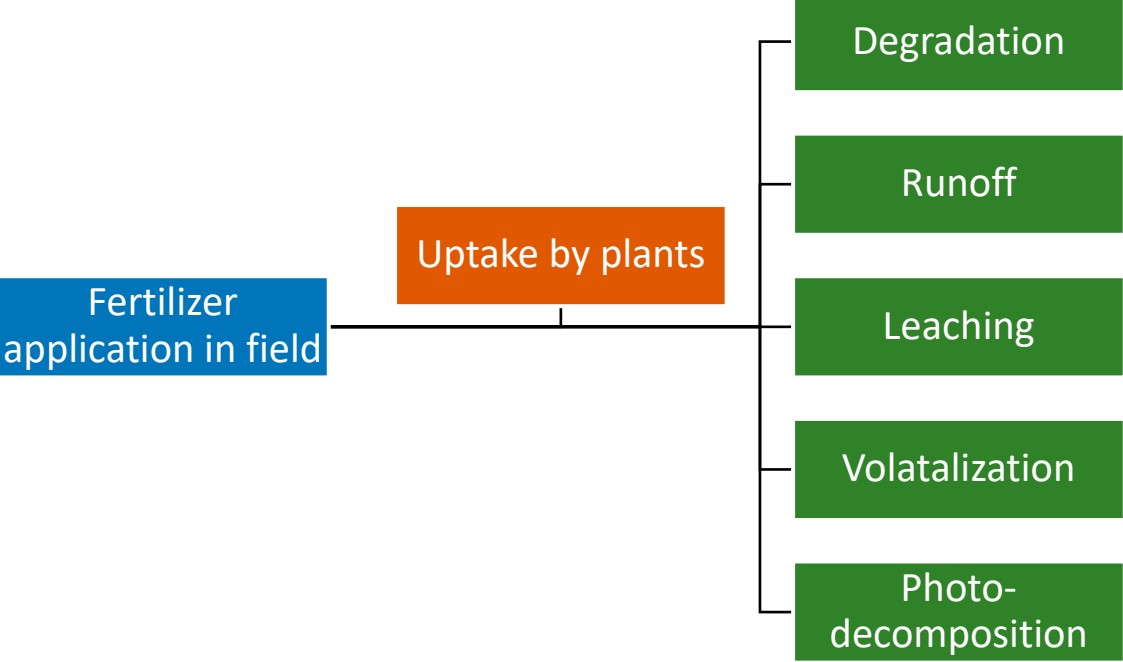

**Figure 1.** Fate of fertilizer application in the field.

Nanofertilizers have emerged as a promising solution to address such challenges, offering higher efficiency and reduced environmental impacts. They can be classified based on their action, nutrient composition, and consistency. These categories include controlled-release nanofertilizers, nanofertilizers for targeted delivery, plant growth-stimulating nanofertilizers, water and nutrient loss-controlling fertilizers, inorganic and organic nanofertilizers, hybrid nanofertilizers, nutrient-loaded nanofertilizers, and various consistency-based nanofertilizers such as surface-coated, synthetic polymer-coated, biological product-coated, and nanocarrier-based nanofertilizers.

Controlled-release fertilizers (CRFs) are promising nanofertilizers with granular structures that deliver nutrients to plants over an extended period, ranging from weeks to months [3]. In addition, controlled-release fertilizers can improve the environmental sustainability of agriculture by reducing the release of nutrients into the environment (Figure 2). Nanomaterials, such as carbon nanotubes, graphene, and quantum dots, have unique properties that make them ideal for controlled-release applications [4]. Their small size, large

surface area–to–volume ratio, and ability to be coated with various materials to control the release rate enhance the efficiency of nutrient delivery. These materials can also improve the granular mechanical strength of fertilizers and provide leaching resistance [5].

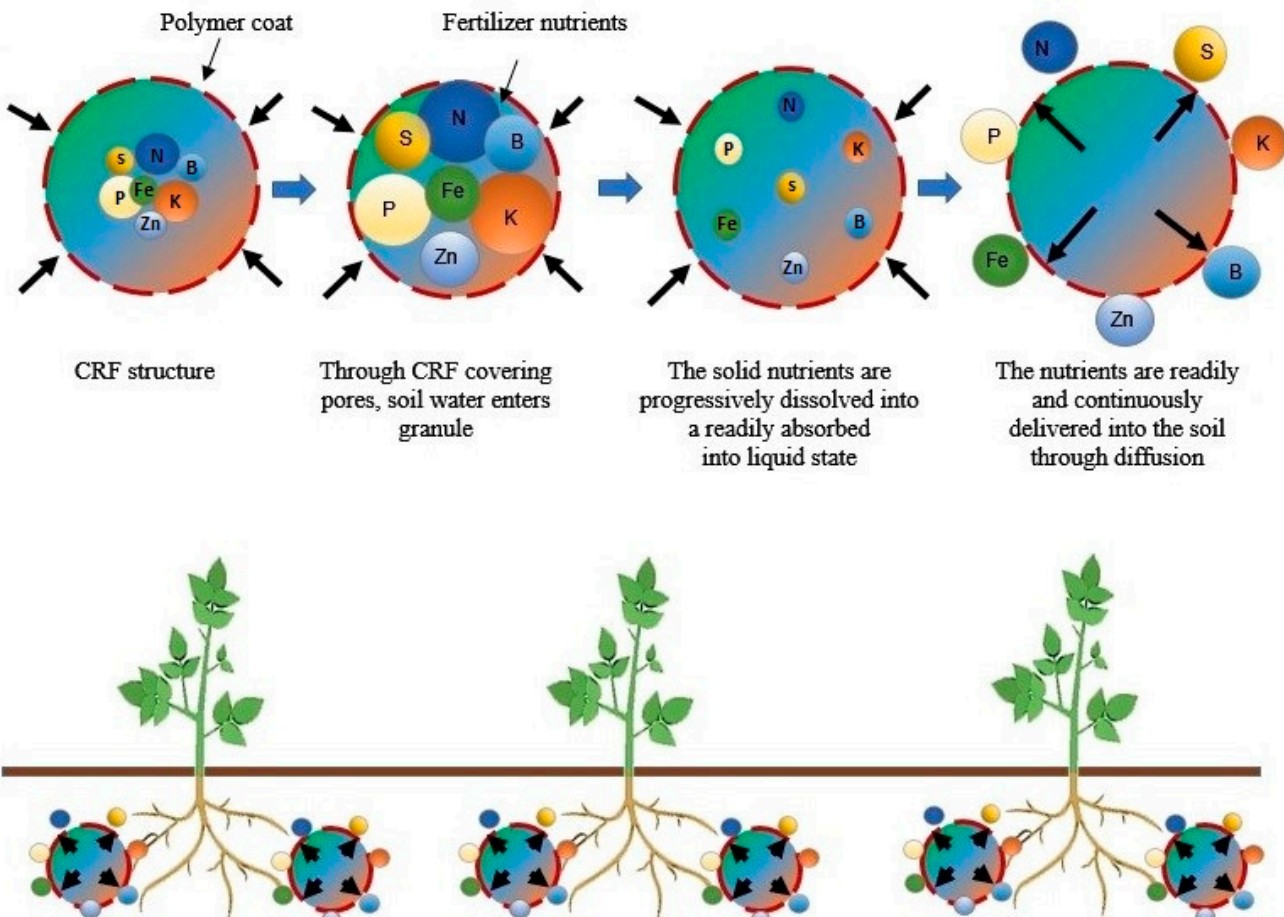

**Figure 2.** Mechanism of action of controlled nutrient release nanofertilizers in the field.

This review article aims to comprehensively describe the various types of nanofertilizers available and their potential applications in modern agriculture. We will discuss their effectiveness, advantages, and disadvantages, along with the materials and strategies for controlled and targeted delivery of nanoparticles (NPs). Furthermore, we will delve into the qualities of an effective nanofertilizer, potential risks related to its application, and the future outlook for this emerging technology. By evaluating and comparing different nanofertilizers, we hope to offer valuable insights for researchers, farmers, and other stakeholders in the agriculture sector.

## 2. Nanofertilizer Types

Nanofertilizers contain nanosized particles, which plants can absorb and improve crop yields. They are the product of a new technology with potential applications in agriculture, but their classification has some inconsistencies. However, some definitions also include other products, such as nanoscale delivery systems and nanobiosensors. The scientific community has been perplexed by the contradictory definitions of nanofertilizers. For instance, nanofertilizers are classified as a subset of nanotechnology and also as a type of fertilizer. This uncertainty has led to a lack of clarity on the definition and categorization of nanofertilizers, which may lead to misunderstanding when debating their use and possible advantages.

Nanofertilizers can also be classified based on the material used. For example, some nanofertilizers are made with carbon nanotubes, while others are made with polymers or

metals. Each type of nanofertilizer has different properties and can have different effects on plants. Here in this review, they are broadly classified based on the nutrients they carry, the actions they perform, and consistency (Figure 3). Understanding the nature of nanofertilizer is essential to find the best application method. Nanofertilizers can be applied to plants through foliar, water, and soil application [6].

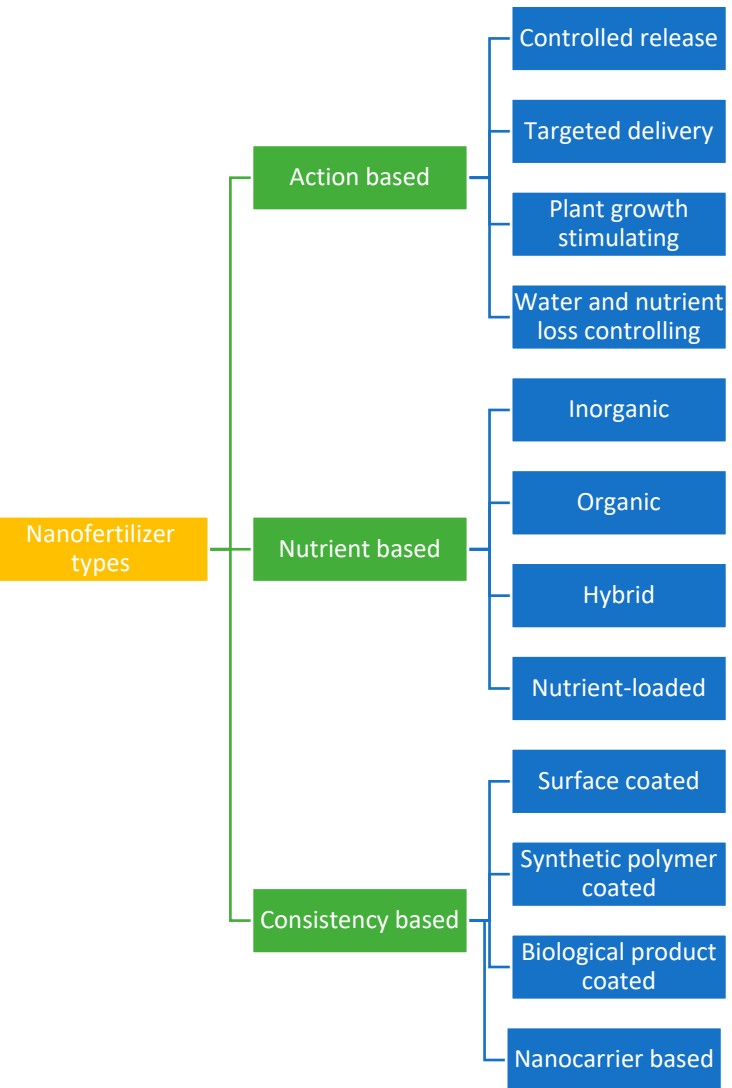

**Figure 3.** Classification of nanofertilizers.

### 2.1. Action-Based

Nanofertilizers can be categorized into five types: action-based, controlled-release, targeted delivery, plant growth-stimulating, and water and nutrient loss controlling. These innovative fertilizers offer a range of benefits, such as improved nutrient utilization, controlled nutrient release, targeted nutrient delivery, enhanced plant growth, and reduced nutrient loss, making them valuable for sustainable agriculture, as described below.

### 2.1.1. Controlled-Release Nanofertilizers

Controlled-release nanofertilizers have emerged as a promising solution for addressing the challenges associated with conventional fertilizers, such as nutrient leaching and inefficient nutrient use [7]. These fertilizers utilize nanoparticles to control the release of nutrients, thereby improving nutrient uptake and reducing environmental impacts [8]. Controlled-release nanofertilizers encapsulate nutrients within nanoscale carrier materials

composed of polymers, lipids, or inorganic substances [7]. The release of nutrients from such carriers can be influenced by environmental factors like temperature, pH, and moisture or by stimuli-responsive mechanisms, such as biodegradation or enzyme-mediated degradation [9].

Controlled-release nanofertilizers offer several advantages that contribute to sustainable agriculture. One of these benefits is improved nutrient use efficiency, as the controlled release of nutrients allows for targeted and sustained delivery, leading to better nutrient uptake by plants and reduced fertilizer application rates [7]. Furthermore, these fertilizers help to reduce the environmental impact of agricultural practices. By minimizing nutrient leaching, controlled-release nanofertilizers decrease the contamination of water bodies and reduce eutrophication, ultimately protecting aquatic ecosystems [8,10]. Additionally, controlled-release nanofertilizers are known to enhance crop productivity. Studies indicate that their application can lead to improved crop yield, nutrient use efficiency, and overall plant health [11]. The potential applications of these fertilizers extend to various crops, including rice, wheat, corn, and soybeans, with promising results in improved crop yield and nutrient use efficiency [7]. Continued research and development efforts are crucial to fully harness the potential of these innovative fertilizers in addressing the increasing global demand for food production.

### Carbon-Based

Carbon is essential for promoting life, as it is found in all organic compounds and involved in biochemical pathways. Furthermore, nanostructured carbon can play vital roles in plants [12], such as biochar, a form of leftover biomass from plants in fields. The biochar contains numerous carbon nanostructures that undergo oxidation upon exposure to air and create pores on the surface. These pores are beneficial as they can absorb micronutrients from the soil to be released in the future, as well as a large amount of water to be used during times of drought, thus helping the soil to remain wet and nourished for seed germination and plant growth [13,14].

The advent of carbon-based nanofertilizers presents a revolutionary solution that holds immense promise, encompassing a multitude of potential advantages, including improved nutrient delivery, uptake, and overall plant growth [15]. Carbon nanotubes (CNTs) are one of the most studied carbon-based nanofertilizers, known for their high aspect ratio, mechanical strength, and unique chemical properties [16]. CNTs' application to plants improves nutrient delivery and uptake, increasing crop productivity [17]. Graphene is another type of carbon-based nanofertilizer, which has shown promise in increasing nutrient use efficiency and promoting plant growth [18].

### Chitosan-Based

Chitosan, a biopolymer, is widely recognized as a leading choice among agriculture, food, and health experts. It is a deacetylated derivative of chitin derived from arthropod shells. Due to its structure, chitosan, a cationic polymer, can interact with negatively charged materials. Chitosan forms complexes with fertilizer molecules, increasing their availability to the plants. Furthermore, it has the added benefit of being adjustable in size. Chitosan NPs can serve as a vehicle for the controlled release of NPK fertilizers in agricultural applications. The process is relatively easy, as the fertilizers can be dissolved in the nanoparticle solution and stirred magnetically [19]. It is easy to modify chitosan molecules to absorb and release plant growth regulators, herbicides, insecticides, and fertilizers [20]. As a carrier and control release matrix, chitosan protects biomolecules from pH fluctuations, light, and temperature extremes and prolongs the release of active substances, safeguarding plant cells from burst release. Chitosan-based nanofertilizers improve the absorption of nutrients by plants [20].

Furthermore, chitosan's ability to form complexes with metal ions helps to reduce its toxicity, making them safer to use. Slow-release nanofertilizers, such as chitosan NPs (CN) and potassium-incorporated chitosan NPs (CNK), are of interest for reducing soil

nutrient losses and preventing land degradation associated with established fertilizer use. These NPs were characterized using Fourier-transform infrared spectroscopy (FTIR), transmission electron microscopy, a field-emission scanning electron microscope, and atomic force microscopy (AFM) techniques [21]. Membrane diffusion studies revealed the slow potassium release property of CNK. Pot trials with *Zea mays* plants demonstrated that soils amended with reduced potassium rates (75% CNK) significantly increased the fresh and dry biomass accumulation by 51 and 47%, respectively, concerning positive control (100% KCl) [21]. Using CNK also improves the physical properties of soil, such as porosity, water conductivity, and friability, favoring root growth and thus allowing plants to uptake higher quantities of nutrients [21]. No harmful effects of the nanoformulation were observed in the study, and the treatments showed better carbon cycling and higher soil microbial activity. It is hypothesized that CNK conditions the soil by cohesively bonding soil particles, stabilizing the soil aggregates by providing a coating, and reducing the disruptive forces that cause degradation [21]. The sustained nutrient release of CNK synchronizes with crop demands, reducing fertilizer requirements and increasing productivity.

Clay-Based

NPs composed of clay possess large surface areas and nanolayer reactivity and can be utilized to fabricate CRF formulations. Nanoclay is a crucial component in CRF synthesis because of the active surface it provides for several physicochemical and biological processes [22,23]. Organoclays are clay minerals with particle sizes less than 2 mm and demonstrate high specific surface area, hydrogel capacity, charge, and crystal-like structure. In addition to their colloidal particle size, all these qualities are advantageous for CRF production [23–26].

Layer Double Hydroxides

Layer double hydroxides (LDH) are two-dimensional layered compounds composed of intercalated anionic materials within an interlayer spacing that enable the regulated release of anions, regardless of their type, to balance the positive charge of the LDH. Furthermore, the composition and synthetic methods used to produce LDH can result in various properties, such as anion exchange and thermal stability [27]. A study proposed a route to creating a hydrotalcite-like layered double hydroxide structure ([Mg-Al]-LDH) for phosphate fertilization, which resulted in a higher phosphorus content than previously reported [28]. Another study used an Mg/Al layered double hydroxide to intercalate three anionic herbicides (2, 4-D, MCPA, and picloram). The resulting complexes were tested for controlled release in water and soil columns, and the results showed that the LDHs could be used in slow-release formulations of acid herbicides [29].

Nanocapsule-Based

Nanocapsules are microscopic capsules typically composed of organic or inorganic material used to encapsulate and deliver fertilizers [30]. They are made from biopolymers, lipids, silica, metal oxides, and carbon nanotubes [31]. The nanocapsules form a protective barrier and release the nutrients slowly over time, allowing for more efficient fertilizer use, greater control over its application, retaining nutrient stability, and providing a controlled release to the crops, thus ensuring that the nutrients are delivered correctly and at the correct times. Additionally, the slow release of nutrients reduces the risk of leaching.

Several studies have demonstrated the potential benefits of nanocapsule-based nanofertilizers in enhancing nutrient uptake and improving crop yields. For instance, it was reported that applying encapsulated zinc oxide NPs to maize plants significantly enhanced photosynthetic rate, growth, and yield under cobalt stress [32]. Furthermore, encapsulated urea nanofertilizers provide more efficient nitrogen release and improve plant growth than conventional urea [31].

### Nanogel-Based

Nanogels are spongy materials composed of a polymer mixed with a liquid, which can absorb and gradually release fertilizers or other substances over time. The nanogels are impregnated with nitrogen, phosphorus, and potassium, indispensable elements for facilitating plant growth [33,34]. Nanogels have applications in fertilizers, drug delivery, and water purification. The nanogel particles can absorb and retain water in the soil, which can help to improve water retention and reduce soil erosion. They also have a high surface area–to–volume ratio, which can contain more nutrients than traditional fertilizers. Furthermore, they are biodegradable and non-toxic and thus are environmentally safe [35]. In a study comparing the effects of different fertilizers on *Abelmoschus esculentus*, the application of a composite fertilizer containing calcium phosphate nanogel and dipotassium hydrogen phosphate resulted in an increased germination rate of *A. esculentus* from 87% to 95%, as well as improved activity of the enzymes amylase, protease, and nitrate reductase, and increased weight per fruit from 59 g to 65 g [36].

### Polyurethane-Based

Polyurethane-based nanofertilizers represent a cutting-edge category of fertilizers that harness the power of polyurethane NPs, which are synthesized from a polymer composed of organic units interconnected by urethane links. The polyurethane-based nanofertilizers are typically produced through ring-opening metathesis polymerization (ROMP), which involves creating a monomer from two different molecules containing a double bond, which is then polymerized. The resulting polymer creates NPs that deliver nutrients to the soil [37]. These NPs can be used to deliver essential nutrients such as nitrogen, phosphorus, and potassium to improve the efficiency of traditional fertilizer products. The polyurethane matrix of such nanofertilizers protects the particles from the environment and provides a controlled release of nutrients into the soil. The controlled release nanofertilizers offer several advantages over traditional fertilizers, including a more efficient and longer-lasting nutrient release, simultaneously releasing multiple types of nutrients, and a higher water-holding capacity that can reduce soil erosion.

Traditional fertilizers are coated with polymers such as polyurethane to engender slow-release fertilizers [38]. Polymeric materials, including cottonseed oil, can be synthesized using cost-effective, biodegradable, and renewable sources. The coating increases surface roughness, reduces surface energy, superhydrophobic qualities of polyurethane prevent water from coming into contact with the fertilizer in its liquid form [39]. However, not all nutrient elements can be incorporated with polyurethane. For example, sulfur's oxidation and brittleness prevent it from being used as a slow-release fertilizer. In such cases, an alternative natural polymer is employed to absorb water. Unfortunately, the nutrient release duration of this product is shorter than 30 days. A novel approach was devised to circumvent this constraint, which entails turning natural polymers into bio-polyols by coating fertilizer with polyurethane from wheat straw and liquefying the wheat straw using solvents such as ethylene glycol/ethylene carbonate. This liquid is then mixed with polymethylene polyphenyl isocyanate and castor oil to create a bio-based polyurethane, providing a more viable solution than synthetic polymers, which are too costly. [40]. Polyurethane-based nanofertilizers are likely to become popular for use in agricultural applications in the future.

### Starch-Based

Starch-based nanofertilizers comprise nanocrystals derived from starch, which can be readily dissolved in water or administered to plants in either liquid or aerosol form [41]. These nanocrystals effectively fertilize crops using a high-quality renewable energy source without creating chemical waste. A study demonstrated the effectiveness of nanotechnology in agriculture by synthesizing a polymeric formulation of polyvinyl alcohol -starch as a substrate for the slow release of Cu-Zn micronutrient-carrying carbon nanofibers (CNFs) [42]. The results showed that the plants treated with the nanofertilizer were sig-

nificantly taller than the control, and the nanofertilizer was also influential in scavenging reactive oxygen species. By contrast, the metal release profile of the nanofertilizer was considerably lower than that of the CNFs [42].

Zeolite-Based

Zeolites are a class of microporous, aluminosilicate minerals that have been used for decades as a soil amendment due to their ability to absorb and retain water, nutrients, and other organic compounds [43]. Developing nanofertilizers based on zeolites involves the synthesis of zeolite NPs, which are then combined with various other compounds to create fertilizer. These nanofertilizers are small, allowing them to penetrate deeper into the soil and deliver nutrients to a crop's root zone more efficiently [44,45]. In addition, zeolites also act as a reservoir for storing and releasing nutrients over time, providing a more consistent supply of nutrients to plants. Furthermore, zeolite-based nanofertilizers can be tailored to the specific nutrient needs of a crop, ensuring that only the required nutrients are provided, thus reducing the cost of fertilization.

Table 1 compares known controlled-release nanofertilizers. The best nanofertilizer for the application depends on soil type, crop needs, and environmental conditions. Overall chitosan-based nanofertilizers balance the advantages of controlled release, biodegradability, and ease of modification, making them a promising candidate for various agricultural applications. Moreover, they can be easily combined with other nutrients and agrochemicals, providing a versatile solution for sustainable agriculture. Regardless, it is crucial to consider the specific needs of the plants and the local environment before choosing the most appropriate nanofertilizer.

**Table 1.** Comparison of various types of controlled-release nanofertilizers.

| Nanofertilizer Type | Advantages | Ref. | Disadvantages | Ref. |
|---|---|---|---|---|
| Carbon-based | promote plant growth, increase water and nutrient retention, help during drought | [46] | time consuming synthesis methods | [47] |
| Chitosan-based | biodegradable, adjustable in size, easy to modify, protect biomolecules from environmental factors | [48,49] | hydrophilicity, weak mechanical properties, low gas permeability, low encapsulation efficiency | [50] |
| Clay-based | large surface area, nanolayer reactivity, regulate the release of anions | [51] | can inhibit leaf growth and transpiration | [52] |
| Nanocapsule-based | controlled nutrient release, efficient nutrient delivery, reduced risk of leaching | [53,54] | require complex synthesis processes, subject to material limitations | |
| Nanogel-based | highly soluble, biodegradable, non-toxic, improves water retention | [55] | limitations regarding the optimization of biodistribution, degradation mechanism, and component toxicity | [56] |
| Polyurethane-based | controlled nutrient release, improved water-holding capacity, reduced soil erosion | [57] | weak chemical and thermal stability, rapid elimination, lower polymer life span due to the formation of acid monomers in polymer matrix | [58] |
| Starch-based | renewable energy source, effective nutrient delivery, minimal chemical waste | [59] | expensive and time-consuming, unstable nature | [60] |
| Zeolite-based | improved nutrient delivery, tailored nutrient provision, reduced fertilization cost | [45,61] | require specific formulations and synthesis processes for optimal results, not useful in the management of anionic nutrients and need to be complemented with biopolymers and biopolymer complexes | [62] |

### 2.1.2. Nanofertilizers for Targeted Delivery

Nanoaptamers

Nanoaptamers belong to an innovative new form of fertilizer delivery system revolutionizing agricultural cultivation. Nanoaptamers target specific molecules in the soil and deliver nutrients or other molecules directly to the plant. They are tiny molecules that can bind to plant hormones and enzymes and deliver them effectively and efficiently to the plant. Nanoaptamers are reported to increase plant nutrient uptake from the soil [63,64]. Aptamers, which are small molecules made of either oligonucleotides or peptides, can be used to modify the surface of nanofertilizers. This modification enables the release of nutrients within the nanostructure once activated by signals emanating from the rhizosphere. Nanoaptamers can deliver fertilizer to plants by attaching fertilizer components to a nanoparticle, such as a gold nanoparticle or a liposome. This approach safeguards the aptamer from degradation and enables its targeted delivery to the intended plant cells. The aptamer can then bind to a specific receptor on the surface of the plant cell, allowing the nanoparticle to enter the cell and release the fertilizer payload.

Nanoaptamers are employed to regulate the dissemination of fertilizer. By forming a connection between the root system and the soil microorganisms, they enable the precise administration of the desired amount of fertilizer. Previous studies have demonstrated the effectiveness of polymer-coated controlled-release fertilizers wherein the aptamer binding to the target causes the polymer to become more permeable, delivering a payload of nutrients [65,66].

Using nanomaterials for the targeted delivery of essential nutrients, pesticides, and genetic materials could significantly enhance the agricultural industry [67–69]. The use of nanoaptamers in agriculture is still a relatively new concept, and much is still to be learned about their potential applications. As nanoaptamer technology develops, we expect to see even more applications for these revolutionary fertilizers soon. Future usage of nanoaptamers as a commercial fertilizer is possible. However, they are not frequently used, and there is insufficient information on the subject.

Others

Other target-based nanofertilizers include nano-coated urea, iron oxide NPs, nano-hydroxyapatite, carbon-based nano-nutrient carriers, nano-emulsions, and nano-encapsulated micronutrients, as well as clay-based nanofertilizers [7,69]. These innovative formulations allow for the controlled release and enhanced uptake of nutrients, resulting in increased crop yields and reduced environmental impacts [31]. However, the functioning and advantages of all of these nanofertilizers overlap with other types.

### 2.1.3. Plant Growth-Stimulating Nanofertilizers

Certain nanofertilizers, such as carbon nanotubes (CNTs), stimulate plant growth by interacting with plant root systems and boosting hormone synthesis. They increase the amount of carbon and other nutrients in the soil. CNTs consist of rolled-up sheets of carbon atoms and possess unique properties that make them excellent fertilizers. CNTs can absorb and release nutrients, improve soil structure, increase water retention, and enhance the growth of plants. In contrast to the unfavorable effects of large-dose fertilizer experiments, low carbon nanotube concentrations may benefit seed germination, root development, and water transport with no phytotoxicity [70]. When used as a fertilizer, the CNTs blend with the soil, releasing plant nutrients. As a soil amendment, CNTs are added directly to the soil to improve its capacity to store nutrients and water. CNTs can penetrate deep into the soil, providing sustenance to the flora over a prolonged period. The diffusion of nutrients over a more extended period inhibits the occurrence of over-fertilization.

Additionally, CNTs can bolster the soil structure, improving its ability to retain water, hindering soil erosion, and augmenting its nutrient storage capacity. CNTs can also help reduce the amount of fertilizer needed, as they can be used in smaller quantities than traditional fertilizers. CNTs have tremendous tensile strength, making them one of the

strongest and smallest-known fibers. They can migrate to systemic areas of plants such as fruits, leaves, and roots, indicating a significant interaction with plant cells [71]. In addition to being employed as a fertilizer or soil amendment, CNTs can also be applied as a protective coating on seeds that aids in preventing pests while simultaneously boosting the seed's capacity to absorb nutrients and water. CNTs have been suggested to enhance the mechanical properties of plant stalks and stem. The high strength and stiffness of CNTs could potentially make these materials more resistant to breaking and collapsing under heavy loads, improving crop yields.

### 2.1.4. Water and Nutrient Loss-Controlling Fertilizers

Nanofertilizers contain NPs capable of controlling the rate at which fertilizers are released into the soil, allowing farmers to use lesser fertilizer while maintaining the same crop output. Several approaches are considered for designing nanofertilizers that can control the release of nutrients and reduce water loss. One method involves encapsulating the nanofertilizer in a porous matrix that can slowly release nutrients over time [72]. Another approach involves modifying the surface of the nanofertilizer to make it hydrophilic, which can increase its water-holding capacity and reduce the amount of water lost to evaporation [73].

Urea coated with NPs of iron oxide, sulfur, calcium, magnesium, zinc, copper, molybdenum, boron, ammonium sulfate, and potassium are some examples of nanofertilizer types that control water nutrient loss in soil [74,75]. Nanobeads and nanoemulsions are two prominent nanofertilizers that control soil water and nutrient loss [76].

### Nanobeads

Nanobeads are minuscule particles engineered to contain slow-release nutrients, diminishing plant water loss [50]. Nanobeads are made from various materials, including iron, carbon, and other metals. The beads are so tiny that they can get into the smallest cracks in the soil and help to fertilize the plants. Nanobeads can also be used to help clean up pollution. One of the most popular nanobead-based fertilizers is commercially available NanoFert, containing macro- and micronutrients [77]. These particles are designed to dissolve in the soil quickly, allowing the nutrients to be taken up by the plant roots. NanoFert is also intended to be low in salt content, so it won't harm beneficial soil microbes or adversely affect the soil structure [77].

Another popular nanobead-based fertilizer, N-Flex from Limagrain Europe, contains nanosized particles that slowly release nutrients over time [78]. The slow release of nutrients helps ensure the plants receive the nutrients they need for optimal growth without the risk of over-fertilization. N-Flex also contains more nitrogen than many traditional fertilizers, which helps promote healthy growth [78].

### Nanoemulsion-Based Fertilizers

Nanoemulsions are tiny droplets designed to contain a mixture of water-soluble and insoluble nutrients. Nanoemulsion-based fertilizers are formulated by combining a surfactant with a liquid and using high-energy mixing to create a stable, homogenous mixture. The surfactant helps to keep the liquid droplets suspended in the water-soluble matrix, allowing them to remain evenly distributed throughout the solution. The droplets are typically smaller than 100 nanometers in diameter, making them much smaller than the particles found in traditional fertilizers. Plants readily absorb and utilize the droplets, resulting in better yields and improved crop health.

A study found that adding 1% paraffin oil nanoemulsion to the Blue-green 11 media significantly increased biomass yield, chlorophyll-a synthesis, cell numbers, $CO_2$ absorption, and biochemical content of the freshwater microalgal strain *Chlorella pyrenoidosa* [79].

Nanoemulsion-based fertilizers are advantageous because their minute droplet size increases plant cell walls' permeability, facilitating plant cells' rapid and complete fertilizer absorption [80]. The capacity of nanoemulsion-based fertilizers to target-specific nutrients

provides an additional advantage. Due to the small size of the droplets, they may be adjusted to provide specific nutrients to plants, allowing farmers to meet the requirements of their crops better. In addition, they can be applied to any conventional irrigation system in various ways. A nanostructured slow-release fertilizer system was formulated in a study by blending nanoemulsions of nano phosphate and potash fertilizer with neem cake and PGPR [81]. Nanoemulsion-based formulations effectively combat fungal pathogens in crops, which is a major cause of crop loss, as demonstrated in multiple research studies, and it can significantly impact crop yields and economic benefits [82,83].

Nanotechnology-based nanoemulsions offer a range of potential applications in agriculture, such as high surface area per unit volume, improved stability, enhanced transparency, and potent rheology. The advantages of spreadability, wettability, bioactivity, and mechanical strength make nanoemulsions especially promising for delivering plant growth-promoting rhizobacteria (PGPRs) in soil [84].

Nanoemulsions are developed by combining two immiscible phases, an aqueous phase and an oil phase, with emulsifiers such as surfactants and co-surfactants. Typical oil phases include captex 355, captex 8000, witepsol, myritol 318, capryol 90, sefsol 218, triacetin, isopropyl myristate, castor oil, and olive oil, while water is used as aqueous phase [85,86]. Emulsifiers play an essential role in nanoemulsion formulation, as they improve kinetic stability [87], interactions [88], and shelf-life [89], as well as reduce the interfacial tension between the two phases. Typical surfactants are cationic, anionic, amphoteric, and nonionic [90]. Scientists are now exploring nanoemulsified microbial-based methods that can reduce the impact of chemicals and provide sustainable solutions to current food and climate challenges. However, additional research is needed to ensure their sustainability and affordability as a strategy for soil health and crop improvement.

### 2.2. Nutrient Based

The increasing global population and consequent demand for food have made it essential to develop sustainable agricultural practices. Nutrient-based nanofertilizers are one such innovation that addresses this challenge by enhancing nutrient availability, uptake, and utilization in plants [7]. This section discusses various nutrient-based nanofertilizers and their benefits in sustainable agriculture.

### 2.2.1. Inorganic Nanofertilizers

Inorganic nanofertilizers include metals, metalloids, and non-metallic NPs, and they can provide essential nutrients, such as nitrogen, phosphorus, and potassium, to plants. These fertilizers are designed to improve the efficiency of nutrient uptake by plants and can be used to improve yields in agriculture [33,91]. The choice of material depends on the desired properties of the fertilizer. Inorganic nanofertilizers offer a unique advantage due to their ability to be customized to the specific nutrient requirements of targeted plants, thereby allowing targeted applications to enhance yield. Inorganic nanofertilizers are already being used in agriculture, and their use is expected to increase as more farmers adopt precision agriculture practices. In a study on safflower plant growth, the foliar application of silicon dioxide NPs (nSiO$_2$) improved canopy spread, stem diameter, plant height, and ground cover [92]. The study suggests that organic fertilizers combined with nSiO$_2$ application can improve safflower production in semi-arid areas.

There is great interest in using inorganic nanofertilizers; several companies are already producing and marketing these products. For better understanding, inorganic nanofertilizers are divided into macronutrient and micronutrient-based nanofertilizers.

### Macronutrient Nanofertilizers

Macronutrient nanofertilizers offer significant advantages for plant growth and environmental sustainability. For example, nitrogen-based nanofertilizers enhance nitrogen utilization efficiency, reducing eutrophication and greenhouse gas emissions [93]. Phosphorous nanofertilizers help plants absorb nutrients and have shown promising results in soil

reclamation [94]. Potassium-based nanofertilizers have higher absorption rates, are more resistant to leaching, and can improve soil physical properties. Calcium-based nanofertilizers contribute to higher crop yields, improved fruit and vegetable quality, and increased disease resistance [95,96]. Magnesium-based nanofertilizers enhance crop growth, quality, and resistance to disease and pests while benefiting various crop types [97]. Lastly, sulfur-based nanofertilizers offer slow-release options for sustained nutrient supply, minimizing the risk of soil acidification [98,99]. The details of each type are described below.

(a) Nitrogen-based

Nitrogen is the most critical nutrient that restricts agricultural production on a global scale. Despite numerous endeavors, the nitrogen utilization efficiency in farming remains less than 50% [100]. In the past few decades, nitrogen over-utilization was done to achieve targeted agricultural yields, a financial and ecological concern of global relevance. [101]. Existing nitrogen fertilizers have a poor utilization efficiency (20%), leading to eutrophication and increased greenhouse gas emissions [102]. Most of the nitrogen in urea is lost owing to rapid volatilization and leaching quickly after application.

Nitrogen nanofertilizers combine nitrogen molecules with NPs such as carbon nanotubes, graphene, and metal oxides. This combination of particles helps increase the available nitrogen in the soil, allowing plants to access more nutrients. The gradual release of nitrogen into the soil from these fertilizers decreases the amount of nitrogen in aquatic systems, thereby reducing the risk of environmental damage from leaching and runoff. One study observed the nutrient release pattern of nitrogen-containing nanofertilizer formulations and demonstrated that nanofertilizer releases nutrients for up to 1200 h, but traditional fertilizer only lasts 300–350 h [103]. Another study suggests using zeolite as a nanofertilizer to increase N usage efficiency [45].

(b) Phosphorous-based

Phosphorous is one of the critical nutrients plants need to grow and thrive. It is an essential component in photosynthesis and helps plants to absorb other nutrients, such as nitrogen and potassium. Phosphorous nanofertilizers are a relatively new type of fertilizer that has the potential to revolutionize the way food is grown. They are more efficient, cost-effective, and environmentally friendly than traditional fertilizers. Using slow-release phosphorus nanofertilizer to supply the crop with phosphorus throughout its life cycle can conserve this element [6]. In an experiment, applying nano-rock phosphate to maize plants resulted in similar phosphorous utilization as superphosphates but at a more affordable cost [104]. Recently the nanoformulations of hydroxyapatite (nHAP; $Ca_{10}(PO_4)_6(OH)_2$) were used to deliver phosphorus to plants [105].

When used in the right amount, it helps plants absorb more phosphorus and other nutrients [106]. Phosphorus nanofertilizers have been demonstrated to be efficacious in soil reclamation processes. A study found that applying rock phosphate-based nanofertilizer enhanced plant growth and yield in degraded soils, particularly when the phosphate-based nanofertilizer was encapsulated in a chitosan shell [107]. A newly developed slow-release phosphorous nanofertilizers stimulated rice growth when applied with chitosan. This nanofertilizer combined poly-beta-amino-esters, graphene oxide, chitosan, poly lactic-co-glycolic acid, and phosphorus in active and barrier layers to control the slow release of phosphorus during the first stages of rice production [108].

(c) Potassium-based

Potassium nanofertilizers, also known as nano potassium, are a modern innovation in agricultural technology. These fertilizers are composed of microscopic particles, enabling them to penetrate deeper into the soil and reach the roots of plants. As a result, they have a higher absorption rate than traditional fertilizers, delivering essential nutrients to plants more quickly and efficiently. Furthermore, nano potassium is more resistant to leaching and is more soluble in water, making it less vulnerable to being washed away by rainfall or

irrigation [109]. These features make potassium-based nanofertilizers significantly more effective in sustaining higher yields over extended periods.

(d)   Calcium-based

Calcium is essential for plants in many ways, including cell division, cell wall formation, and the transport of water and nutrients. Calcium ions are a secondary messenger in signal transduction under various stress circumstances [110]. The improvement of calcium ion levels in the cytosol due to stress signals is predicted by calcium ion-binding proteins, which cause changes in gene expression and plant acclimation to stress circumstances [111,112]. Nitric oxide (NO) was found to promote the concentration of calcium ions in the cytosol under diverse biotic and abiotic stress situations; consequently, calcium ions result in the synthesis of nitric oxide [113]. Research has indicated that using silver nanomaterials on rice roots can impact various plant processes such as responsive protein regulation, calcium ion signaling, transcription, protein degradation, oxidative stress response pathways, cell wall formation, and cell division. The effects of these nanomaterials can vary depending on the concentration used. These findings suggest that NPs can affect plant growth and development by influencing molecular mechanisms and signaling pathways.

Calcium is also necessary for forming seeds and fruit and helps protect plants from disease and pests. However, calcium is not always readily available to plants and can be challenging to apply in the right amounts. Traditional fertilizers that contain calcium, such as lime and gypsum, are often not as effective as they could be.

Multiple variants of calcium-based nanofertilizers have been developed. Some are made from calcium carbonate [114], while others are made from calcium nitrate dope in calcium phosphate [115]. Calcium-based nanofertilizers are effective at increasing the growth and yield of crops. They have also been shown to improve the quality of fruits and vegetables and increase plants' resistance to disease and pests. In recent years, calcium phosphate NPs (CaP) have received significant attention as potential macronutrient nanofertilizers with better nutrient-use efficiency than traditional fertilizers. Their high content of macronutrients, such as phosphorus, and slow solubility in water make them useful as slow-release P nanofertilizers [116].

Additionally, their large surface area can be modified to hold other macronutrient-containing substances, such as urea or nitrate, to create nanofertilizers with improved nitrogen-releasing properties. Studies have shown that CaP nanofertilizers are more effective in agriculture than traditional fertilizers [116]. Biomimetic calcium phosphate NPs (CaP) have been used in biomedicine due to their biodegradability and biocompatibility. Despite this, less progress has been made in precision agriculture for the controlled delivery of active species to plants. A study reports a straightforward and green synthetic method to dope CaP with potassium and nitrogen to create multi-nutrient nanofertilizers (nanoU-NPK) that provide a slow and gradual release of essential plant macronutrients (NPK) and can be used to reduce the amount of nitrogen supplied to durum wheat by 40% concerning conventional treatment, without affecting the final kernel weight per plant [117]. Applying slow-release NPK nanofertilizers is a promising strategy for enhancing fertilization efficiency in precision agriculture [117].

Bio-inspired synthetic calcium phosphate NPs are emergent materials for sustainable applications in agriculture. These salts have self-inhibiting dissolution processes under saturated aqueous media, the control of which is not fully understood [118]. Comprehending the mechanisms involved in the dissolution of particles holds tremendous significance for effectively supplying macronutrients to plants and adopting a valuable synthesis-by-design strategy. Additionally, it bears relevance to the (de)mineralization of bones. In this study, authors shed light on the role of size, morphology, and crystallinity in the dissolution behavior of CaP NPs and on their nitrate doping for potential use as (P and N)-nanofertilizers. They found that morphology actively directs the dissolution kinetics [118].

Amorphous NPs manifest a rapid loss of nitrates governed by surface chemistry. NPs show slower release, paralleling $Ca^{2+}$ ions, supporting detectable nitrate incorporation in the apatite structure and dissolution from the core basal faces [115]. In a further study

on nitrate-doped amorphous calcium phosphate NPs with urea, the same authors found that the process leads to high levels of nitrogen payloads, is cost-effective to scale up, and slows down the release of urea. In tests on cucumber plants, authors found that these NPs promote growth and biomass formation using less nitrogen than traditional fertilizers, making them a promising option for sustainable use as a nanofertilizer [119].

(e)    Magnesium-based

Magnesium is an essential plant mineral in many vital processes, such as photosynthesis, enzyme activation, and protein synthesis. It is also a key component of chlorophyll, the pigment that gives plants green color and allows them to capture sunlight for photosynthesis. However, magnesium is not always readily available to plants and can be challenging to apply correctly. Traditional fertilizers that contain magnesium, such as dolomitic lime and Epsom salts, are often not as effective as they could be.

Several different types of magnesium-based nanofertilizers have been developed. Some are made from magnesium oxide, while others are from magnesium sulfate [120]. Magnesium-based nanofertilizers are effective in increasing the growth and yield of crops. Magnesium enhances the nutritional quality of fruits and vegetables while concurrently bolstering plants' resilience against diseases and pests. These fertilizers benefit crops such as rice, sugarcane, tomato, and potato. One study considered using nano and common forms of iron and magnesium as foliar applications on black-eyed peas [121]. A factorial experiment with three replicates was conducted in a study using different concentrations of Fe and Mg. The elements were applied to the leaves 56 and 72 days after planting, and data were collected after 85 days. The results showed that Fe significantly affected yield, leaf Fe content, stem Mg content, plasma membrane stability, and chlorophyll content. The most significant effect was observed with two combinations of treatments: $0.5$ g $L^{-1}$ standard Fe + 0.5% nano-Mg and $0.5$ g $L^{-1}$ typical Fe + $0.5$ g $L^{-1}$ standard Mg. Almost all analyzed traits were improved by the foliar application of these two elements, likely due to more efficient photosynthesis [121].

Another study investigated the impact of chitosan and magnesium-nano fertilizers on sesame plants' photosynthetic pigments, protein, proline, and soluble sugar content under drought stress [122]. The results show that chitosan foliar application improved the mean traits of chlorophyll a, b, total, carotenoid, protein, proline, and soluble sugar. By contrast, severe drought stress and no nanofertilizer application decreased chlorophyll content and plant damage. The findings suggest that co-applying chitosan and Mg nanofertilizers could effectively reduce plant damage due to drought stress [122].

(f)    Sulfur-based

Sulfur is an essential plant mineral involved in many vital processes, such as protein synthesis, enzyme activation, and the production of vitamins and hormones [123]. "Sulfur plays a crucial role in activating plant defense mechanisms, aiding in protecting plants against diseases and pests. However, sulfur is not always readily available to plants and can be challenging to apply in the right amounts. Traditional fertilizers containing sulfur, such as sulfur-coated urea, are often not as effective as a sole coating for urea granules and require additional materials for proper application, which can be costly. Additionally, sulfur shells left in the soil can cause an excess build-up and acidify the soil if not properly integrated [124,125].

Several different types of sulfur-based nanofertilizers have been developed [31]. Some are made from elemental sulfur, while others are made from sulfur compounds such as sulfur-coated urea or sulfur-coated potassium sulfate [126]. Sulfur nano-coated fertilizer materials are beneficial slow-release fertilizers as they provide both primary nutrient elements and sulfur. The stability of the coating slows the dissolution rate, resulting in a sustained release of the fertilizer.

Micronutrient Nanofertilizers

Micronutrient-based nanofertilizers are advanced agricultural inputs that use NPs to deliver essential micronutrients to plants more effectively than conventional fertilizers. Micronutrients, such as boron, copper, iron, nickel, zinc, and titanium, are required in small quantities but play crucial roles in plant growth, development, and overall health. Micronutrient nanofertilizers offer numerous advantages to plants, including enhanced nutrient uptake, increased yield, and improved resistance to biotic and abiotic stresses, as discussed below.

(a)    Boron-based

Boron is a micronutrient necessary for optimal plant growth but is often lacking in soils. Boron-based nanofertilizers address this deficiency, providing a more concentrated and targeted boron delivery to crops. Boron-based nanofertilizers combine borate with other materials, such as humic acid, to create NPs. They are then suspended in a liquid or solid form, allowing them to be applied to the soil or the crop as a foliar spray. The boron-based nanofertilizers can penetrate plant cells, providing them with essential micronutrients [127]. A study found that foliar application of low doses of nanofertilizers of zinc and boron could increase pomegranate fruit yield and quality. Applying nano-Zn chelate fertilizer and nano-B chelate fertilizer before full bloom increased leaf concentrations of both microelements [128].

(b)    Copper-based

Copper nanofertilizers are incredibly effective at delivering nutrients to plants. The tiny copper particles can penetrate the plant cells and provide the necessary nutrients to the root system. Direct delivery ensures that the nutrients are absorbed quickly and efficiently. Copper nanofertilizers are incredibly safe to use. They have been thoroughly tested and are non-toxic to humans and animals. In addition, they are not prone to leaching or runoff, making them an excellent choice for sustainable agriculture.

Studies have shown that copper nanofertilizers can increase the yield of wheat, maize, and other crops [129]. They can also help plants to resist pests and diseases, as copper is a natural antimicrobial and antifungal agent [130]. Nevertheless, copper nanofertilizers are much cheaper than traditional fertilizers and can be applied quickly and easily. The affordability of these options makes them particularly appealing to small-scale farmers who lack the financial resources to invest in costlier alternatives.

(c)    Iron-based

Iron is an essential micronutrient for plant growth but is often limited in soils due to low solubility. Iron nanofertilizers are designed to increase the bioavailability of iron in plants. Iron nanofertilizers come in various forms, including nanosized iron particles, nanoencapsulation iron, and nanocomposites [131,132]. Nanosized iron particles are the simplest iron nanofertilizers, typically iron oxide or sulfide. These particles are small enough to penetrate the soil surface, allowing them to be absorbed by plants more efficiently than traditional iron fertilizers. Nanoencapsulated iron comprises particles encapsulated in a protective coating, such as a biopolymer or lipid. The coating helps to protect the iron particles from oxidation, allowing them to remain in the soil for extended periods. Nanocomposites combine iron particles and other materials, such as zeolites, clay, or humic acids. These nanocomposites are formulated to increase plant iron availability by providing an adsorptive or catalytic surface for improved iron absorption. Iron nanofertilizers can improve crop yields on agricultural land, restore iron-deficient soils, and improve water quality in aquatic environments [133]. Iron nanofertilizers can also remediate contaminated soils and reduce acid rain effects [134].

(d)    Nickel-based

Nickel is an essential micronutrient for plant growth and development and one of the most critical elements in agriculture [135]. Nickel is necessary for various metabolic

processes, including photosynthesis, respiration, and nitrogen metabolism. Furthermore, nickel is essential for the production of certain enzymes and hormones, as well as for forming chlorophyll [136]. Plants cannot fully utilize other nutrients without adequate nickel levels, leading to stunted growth and reduced yields. Nickel nanofertilizers are now available in various formulations, including liquid, powder, and granular products. In one study, applying low concentrations of 5 nanometer-sized nickel NPs to 10-day-old wheat seedlings at 0.01 and 0.1 mg $L^{-1}$ caused an increase in the chlorophyll a and b content after exposure to 0.01 mg $L^{-1}$ of nickel NPs [137].

(e)  Titanium-based

Due to the antimicrobial properties, titanium-based NPs (TiO$_2$NP) provide long-term protection against pests and diseases. In a study, these NPs effectively reduced the soil salinity when applied to broad bean crops in the field. [138]. Compared to plant development in alkaline conditions, 0.01% nTiO$_2$ significantly increased plant shoot and root length, leaf surface area, and dry weight under normal conditions [133]. When nanoscale TiO$_2$NP was applied to canola seeds, it boosted seedling vigor and germination, with the highest and lowest germination rates of 2000 and 1500 mg $L^{-1}$, respectively [139]. Compared to the control, greater concentrations of TiO$_2$NP suggested bigger plumule and radical sizes [139]. Similarly, the impacts of various TiO$_2$NP concentrations stimulated wheat seed germination and demonstrated that high concentrations inhibit wheat germination [140].

(f)  Zinc-based

Zinc is an essential component of various plant enzymes that accelerate metabolic activities. Without certain enzymes in plant tissue, plant growth and development would cease, and the creation of carbohydrates, proteins, and chlorophyll would be significantly reduced in zinc-deficient plants. Applying zinc oxide nanofertilizers to soil enhances zinc availability to plants and thus improves crop yields. Zinc oxide nanofertilizers were also found to reduce the leaching of zinc from the soil, thus decreasing the risk of environmental contamination [141]. In addition, studies have indicated that zinc oxide nanofertilizers may improve plant resistance to abiotic and biotic stresses [142–144]. Zinc oxide NPs have been reported to enhance the growth and yield of sesame [145], barley [146], and pearl millet [147]. In a recent experiment, 20 mg $L^{-1}$ ZnO NPs significantly increased the biochemical content and various growth indices of *Citrus aurantium* fruit [148].

Despite these potential benefits, there are still some limitations to zinc oxide nanofertilizers. For example, applying zinc oxide nanofertilizers to soil may lead to an accumulation of zinc in the environment, which can be toxic to plants and animals [149]. Furthermore, there is a lack of research on the long-term effects of zinc oxide nanofertilizers on soil, plants, and the environment.

2.2.2. Organic Nanofertilizers

Organic nanofertilizers are composed of NPs derived from organic materials designed to gradually supply nutrients to the soil. They are eco-friendly, derived from natural sources, and may maintain soil moisture and alter pH levels, enabling plants to acquire essential nutrients more efficiently. Organic NPs are synthesized from atoms or molecules into various forms, such as capsules, polymer conjugates, vesicles, micelles, liposomes, polymersomes, dendrimers, and polymeric NPs [150]. Natural polymers such as gum, xanthan gum, guar gum, gellan gum, mucuna gum, gum gopal, karaya gum, seed polysaccharide, tamarind seed polysaccharide, *Mimosa pudica* seed mucilage, *Leucaena leucocephala* seed polysaccharide, bio-derived materials, chitosan, carrageenan, pectin, and modified clays can be used in nanofertilizer preparation [151].

Organic nanofertilizers may also benefit plant development since the delayed release of nutrients allows for more effective absorption and use, leading to improved growth and yields. Adding organic nanofertilizers can significantly improve soil structure by increasing aggregation and porosity, enhancing water infiltration, aeration, and root penetration [152]. Moreover, organic matter serves as a source of nutrients for soil microorganisms, stimulat-

ing their growth and activity [153]. These microorganisms, in turn, contribute to nutrient cycling and availability, further improving soil fertility and plant growth [154]. Organic matter from plant waste, manure, and compost is used to prepare organic nanofertilizers. In particular, NanoMax-NPK is a commercially available nanofertilizer for all crops thanks to its formulation with organic micronutrients/trace elements, vitamins, and probiotics, as well as its inclusion of multiple organic acids (protein-lacto-gluconates) containing chelated N, P, K, O, amino acids, and organic carbon [150]. Ferbanat and nanonat liquid fertilizers are a nanobiostimulator of a new generation created and manufactured from organic resources. Ferbanat typically includes amino acids, vitamins, micro humates, various biological substances, microelements, and soil microflora to boost the vitality of goods, life processes, and root zone activities [155].

### 2.2.3. Hybrid Nanofertilizers

Nanotechnology-based fertilizers, composed of NPs of a particular nutrient, have been developed to improve the efficiency of traditional fertilizers [69]. Coating the NPs with a protective layer allows their release to be controlled, increasing their availability to plants. Hybrid nanofertilizers combine conventional and nanotechnology-based fertilizers, providing a slow and sustained release of nutrients and improved access to the nutrients [156]. Such nanofertilizers improve fertilizer efficiency and reduce the environmental impact of fertilizer production and use [157]. Nanocomposites and hybrid materials consist of a continuous (polymer) phase and a dispersed (nanofiller) phase that dissipate nanomaterial in minute quantities [158]. Due to the combination of a polymeric matrix and an inorganic nanomaterial, nanocomposites outperform conventional fertilizers in various aspects. The better performance of these materials is a result of their enhanced physical and mechanical properties, such as increased strength, toughness, and rigidity, higher pH tolerance, greater storage stability, heat distortion, and break elongation, improved electrical and thermal conductivity, superior flame resistance, and a higher barrier to moisture and gases [57]. Due to their many design choices, nanocomposites also provide significant benefits for developing functional materials with the characteristics required for particular applications (Figure 4). Hybrid nanofertilizers can offer several potential benefits for agriculture and horticulture. They can also reduce plants' fertilizer needs, potentially leading to cost savings for farmers [159]. Combining the properties of conventional and nanotech-based fertilizers can provide improved nutrient availability and control over nutrient release. Such nano-sized fertilizer particles are injected into irrigation systems, a process called nano-fertigation, which allows for the precise application of fertilizers directly to the roots of plants, leading to improved nutrient uptake and reduced waste.

There are several types of hybrid nanofertilizers, each with its unique properties and benefits. These include nano-encapsulated and nano-clay fertilizers, which have already been discussed under the action-based fertilizers category. In addition, nanobiofertilizers, which combine the benefits of nano-size fertilizer particles with beneficial microorganisms, are also gaining popularity. These microorganisms help to improve soil health and increase the efficiency of nutrient uptake by plants through various mechanisms (Section 2.3.3 (b)).

### 2.2.4. Nutrient-Loaded Nanofertilizers

Nano-porous zeolites are nutrient-loaded nanofertilizers that can effectively provide crops with nutrients. These minerals are formed by a reaction between volcanic ash and alkaline lake water, resulting in a honeycomb-like structure that offers a high degree of porosity, facilitating the absorption and retention of moisture, nutrients, and other compounds, thus making zeolite fertilizers an excellent choice for soil amendment and fertilization. To create a nanofertilizer tailored to different crop needs, the natural zeolite is ground into a powder mixed with nitrogen, phosphorus, and potassium, increasing the zeolite particles' surface area and improving the soil's ability to absorb and retain nutrients. Engineered nano-porous zeolites, also known as aluminosilicates, are distinguished by the presence of micro- (2 nm), meso- (2–50 nm), and macropores (>50 nm) that may be

used as fertilizer carriers owing to their increased ion-exchange and adsorption capacities. Nano-porous zeolite fertilizers improve soil fertility, making it more suitable for crop growth [44]. Furthermore, the zeolite particles' water retention capability helps reduce the need for frequent irrigation [44], which can benefit areas where water resources are scarce or expensive. Additionally, zeolite fertilizers can help to reduce soil erosion due to the increased water retention capabilities of the zeolite particles.

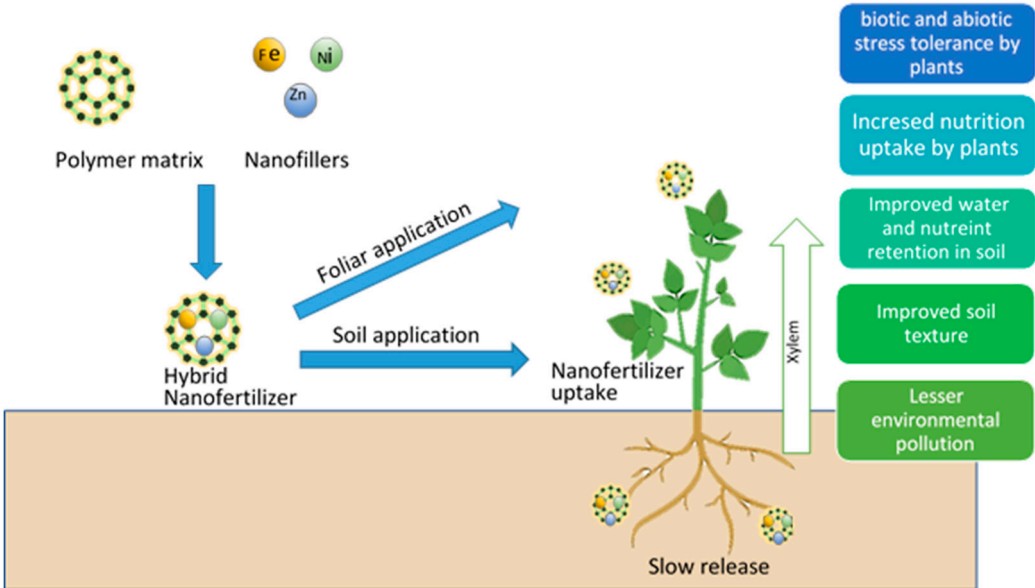

**Figure 4.** Mechanism of action of hybrid nanofertilizers in the field.

### 2.3. Consistency-Based Nanofertilizers

Consistency-based nanofertilizers include surface-coated and nanocarrier-based technologies, providing a promising solution for sustainable agriculture, offering benefits such as improved nutrient utilization, reduced nutrient loss, and minimized environmental impact [7]. Surface-coated nanofertilizers consist of synthetic polymer-coated and biological product-coated NPs [160]. Synthetic polymers, such as polyacrylamide, can increase nitrogen use efficiency [93], while biological product coatings, such as alginate, can enhance phosphorus availability and uptake [161]. Nanocarrier-based nanofertilizers, which utilize carriers such as mesoporous silica NPs, can improve nutrient solubility, stability, and availability, increasing crop yields [162].

#### 2.3.1. Surface-Coated Nanofertilizers

Surface-coated nanofertilizers comprise NPs of silica, iron, and other nutrient-containing materials, including established fertilizers with nanoscale additives, nanoscale coatings, and fertilizers coated with nanoscale materials [163]. Surface-coated nanofertilizers are made by coating fertilizer particles with nanomaterials such as gold, silver, carbon, and titanium dioxide. The coating facilitates the adherence of fertilizer particles to plant surfaces and their penetration into plant cells. This, in turn, enhances the absorption rate of the fertilizer, leading to greater efficiency in its utilization and improved plant growth.

#### 2.3.2. Synthetic Polymer-Coated

While nanofertilizers are known to be more efficient in delivering nutrients to plants, they can also be challenging to handle due to the small size. To address the issue, researchers have developed synthetic polymer-coated nanofertilizers. Such nanofertilizers contain a thin coating of synthetic polymer. The coating helps to protect the nanofertilizers from environmental degradation and facilitates better handling [164,165]. It also helps to ensure that the nanofertilizers are evenly dispersed throughout the soil so nutrients can be delivered

more efficiently and evenly to the plants. Fertilizers with polymer coatings are appropriate for high-value applications because they prevent nutrient loss. These fertilizers have a nitrogen release pattern that is more complex. The polymer matrix diffusion continues over time, and the consistency of matrix regulates how quickly nutrients are released.

Additionally, the nutrient release is influenced by the coating thickness, chemical makeup, temperature, and moisture. The temperature and humidity of the root zone control the pace at which nutrients are released from controlled-release fertilizers. Some promoted products are Nutricot, Osmocot, and Polyon [166].

One example of a synthetic polymer-coated nanofertilizer is polyethylene-coated urea, made by coating urea particles with a thin layer of polyethylene. Polyethylene coating protects urea particles from environmental degradation and facilitates handling [167]. Additionally, the polyethylene coating helps keep the urea particles evenly dispersed throughout the soil, delivering them more evenly to the plants. Another example of a synthetic polymer-coated nanofertilizer is polyvinyl chloride-coated zeolite, made by coating zeolite particles with a thin layer of polyvinyl chloride. The polyvinyl chloride coating helps protect the zeolite particles from environmental degradation and facilitates their handling [168].

### 2.3.3. Biological Product-Coated

(a)    Organic compound-coated

The nanofertilizer coating helps to improve the uptake, transport, and availability of essential nutrients in the soil, resulting in more efficient and adequate fertilization [7]. Biological product-coated nanofertilizers offer numerous benefits over traditional fertilizer applications [69] and are composed of organic materials, such as humic acid and NPs [169]. Nanofertilizers coated with biological products benefit organic farming by boosting soil fertility, improving water retention, and activating beneficial microbial activity. Nanofertilizers coated with plant growth regulators enable plants to provide critical nutrients while regulating their growth and development.

Clay-based nanofertilizers, characterized by their small, round particles, are the most widely used in the market due to their ease of application and even coverage on plant surfaces. Additionally, clay coating seems more effective in delivering nutrients to the plant, meaning less fertilizer is needed to achieve the desired results [170]. The nutrient release pattern of nanofertilizer carrying nitrogen show that nano-clay-based fertilizer formulations (zeolite and montmorillonite with a dimension of 30–40 nm) are capable of releasing nitrogen for a more extended period (>1000 h) than conventional fertilizers (500 h) [103].

A study found that oligo-alginate, a biopolymer, can improve plant tolerance to abiotic stress such as salinity. The most potent effect on plant growth was the 32,000 g mol$^{-1}$ fraction of the oligomer. Plants treated with the oligo-alginate and exposed to salt stress showed less negative impact than those not treated, with smaller reductions in weight, pigment content, and antioxidant activity, thus indicating that low molecular mass oligo-alginate can be used as a plant biostimulator to limit the adverse effects of salinity [171]. It is essential to apply whichever type of biological product-coated nanofertilizer is chosen correctly and per the manufacturer's instructions to maximize its potential benefits and prevent possible harm to plants.

(b)    Microbe-coated (Nanobiofertilizers)

Nanobiofertilizers combine NPs and beneficial microorganisms, improving plant nutrient availability, uptake, and use efficiency. NPs, due to their small size and high surface area, can effectively transport nutrients to plant roots and tissues, while the microorganisms help in the solubilization, mineralization, and fixation of nutrients [31]. The organic constituents of nanobiofertilizers play a vital role in essential processes such as nitrogen fixation, phosphate solubilization, and soil nutrient replenishment [172]. In addition, the nanoparticle coating, which incorporates nanomaterials such as chitosan, zeolite, and polymers, ensures a consistent release of nutrients to crops [173]. The unique properties of

these nanoparticle-coated biofertilizers, including increased surface area, nano-size, and reactivity, facilitate proper nutrient acquisition for crop fertilization and the sustainable delivery of bioavailable nutrients to plants.

A study on the effects of vermicompost, nanobiofertilizer, and biochar on *Echinacea purpurea* L. growth found that vermicompost and nanobiofertilizer combination treatments resulted in the highest fresh and dry root weights of the plant [174]. Another study developed a lightweight nanocomposite biofertilizer composed of acylated homoserine lactone-coated Fe-carbon nanofibers and bacterial endospores in activated carbon beads [175]. When tested on leguminous and non-leguminous plants, the nanofertilizer significantly increased biomass, root length, chlorophyll, and protein contents after 30 days [175]. Several studies show that Nanobiofertilizers have improved the growth of cereal crops such as maize, legume crops like chickpea, pigeon pea, snap bean, and soya bean; horticultural crops such as tomato, cucurbits, potato, and apple; and forage crops such red clover and sorghum [176].

Nanoparticle application with microorganisms for nanobiofertilizer manufacture is considered safe. In a study on the impact of gold NPs on selected PGPR, no effect was observed on *Pseudomonas putida*, while a significant growth increase was observed in *P. fluorescens*, *Paenibacillus elgii*, and *Bacillus subtilis* [177]. NPs have been found to improve plant growth and crop production when combined with other biofertilizers. In particular, iron (III) oxide ($Fe_2O_3$) NPs, when combined with flower waste and cow dung, have shown significant enhancement in the growth of tomato plants [178].

### 2.3.4. Nanocarrier-Based Nanofertilizers

Nanocarrier-based nanofertilizers transport and deliver nutrients to soil and plants. These nanofertilizers can increase nutrient availability, reduce nutrient losses, and enhance plant growth. To design new generation nanocarriers for the biofortification of nutrients, controlled release capabilities for effective nutrient dispersal and mitigation of harmful effects on the environment and human health is a prerequisite. Nanocarrier-based nanofertilizers offer benefits such as controlled nutrient release, enhanced nutrient uptake, and reduced environmental impact [7,8,11]. Applications in various crops have shown improved yield and nutrient use efficiency [7]. Nanofibers, polymer–nanocellulose–clay composites, silk-fibroin-derived nanocarriers, and carboxymethyl cellulose are currently used nanocarriers [179].

Each nanofertilizer type provides unique advantages, including targeted delivery, enhanced nutrient uptake, and improved retention in soil (Table 2). However, there are potential drawbacks, including cost, environmental concerns, and a lack of comprehensive understanding of long-term impacts. Hence, when selecting a nanofertilizer type, it is essential to meticulously consider these factors in conjunction with specific agricultural requirements and contextual considerations.

**Table 2.** Advantages and disadvantages of each nanofertilizer type.

| Nanofertilizer Type | Advantage | Reference | Disadvantages | References |
|---|---|---|---|---|
| Controlled-release nanofertilizers (CRNFs) | Gradual release of nutrients, reducing nutrient leaching and losses to the environment | [7] | More complex manufacturing processes, potentially increase production costs | [8] |
| | Improved nutrient use efficiency, resulting in higher crop yields | [31] | Limited availability and high cost may hinder widespread adoption | [157] |

**Table 2.** *Cont.*

| Nanofertilizer Type | Advantage | Reference | Disadvantages | References |
|---|---|---|---|---|
| Nanofertilizers for targeted delivery | Precise delivery of nutrients to specific plant tissues, enhancing nutrient uptake | [69] | Potential risks to non-target organisms due to high specificity | [69] |
| | Reduced application rates, minimized environmental impact and conserving resources | [180] | Further research is needed to fully understand the long-term effects on soil health and ecosystems | [181] |
| Plant growth-stimulating nanofertilizers (PGSNFs) | Enhanced plant growth, leading to increased crop yields | [182] | Possible unintended effects on plant physiology and gene expression | [183] |
| | Reduced dependency on chemical fertilizers, lower environmental pollution | [173] | Long-term impacts on plant health and soil ecosystems not fully understood | [8] |
| Water and nutrient loss-controlling fertilizers | Improved water use efficiency, reducing irrigation requirements | [69,184] | Limited research on the long-term impacts of WNLCFs on soil health | [157] |
| | Prevention of nutrient leaching, minimized environmental pollution | [8,31,150,157, 185] | Potential for increased production costs due to more complex formulations | [186] |

## 3. Materials and Strategies for the Controlled and Targeted Delivery of NPs

Generally, nanofertilizers comprise tiny particles of metal oxides, such as titanium dioxide or zinc oxide, or polymers, such as polyethylene or polystyrene. Each of these materials has different properties that make them ideal for different types of fertilizer delivery. For example, titanium dioxide and zinc oxide are highly absorbent and can deliver nutrients directly to plant cells. Polymers, conversely, are more flexible and can be designed to release nutrients over time, allowing for more precise and controlled delivery.

Nanofertilizers are directly applied to plants through spraying, drenching, and coating [187] or indirectly through soil application [188]. Spraying is the most common application method in agriculture, as it is relatively easy to apply and can be done quickly. Nevertheless, it is crucial to acknowledge that nanofertilizers can be readily washed away by rainfall or irrigation due to their diminutive size. Therefore, it is imperative to apply them in a manner that guarantees the retention of particles within the soil.. Drenching and coating are other standard methods of delivery. These can control the release of nutrients over time and ensure that the fertilizer is evenly distributed throughout the soil (Figure 5).

When NPs are applied to the leaves, they must pass through the waxy cuticle, a natural barrier that prevents water loss and protects the leaf. Nevertheless, NPs within the range of 0.6 to 4.8 nm can still permeate the leaf through hydrophilic and lipophilic routes, contingent upon the polarity or non-polarity of the solute. NPs larger than 5 nm have been observed to reach the leaf without a clear explanation [188]. Studies have used transmission electron microscopy and confocal laser scanning microscopy to investigate how plant species take up NPs. The results show that NPs penetrate the leaf and are transported to the root through the vascular system. The phloem vessels, which carry photosynthates and other compounds from the leaves to the roots, are responsible for transporting NPs to the root.

The root system of higher plants can also absorb NPs through pores. Once inside the root, NPs penetrate the cell wall and enter the intercellular space. Alternatively, NPs can be absorbed by a cell-to-cell transfer channel, where they enter the cell cytoplasm and permeate it. These NPs get transferred to neighboring cells through plasmodesmata. Only xylem channels are involved in the movement of NPs between roots.

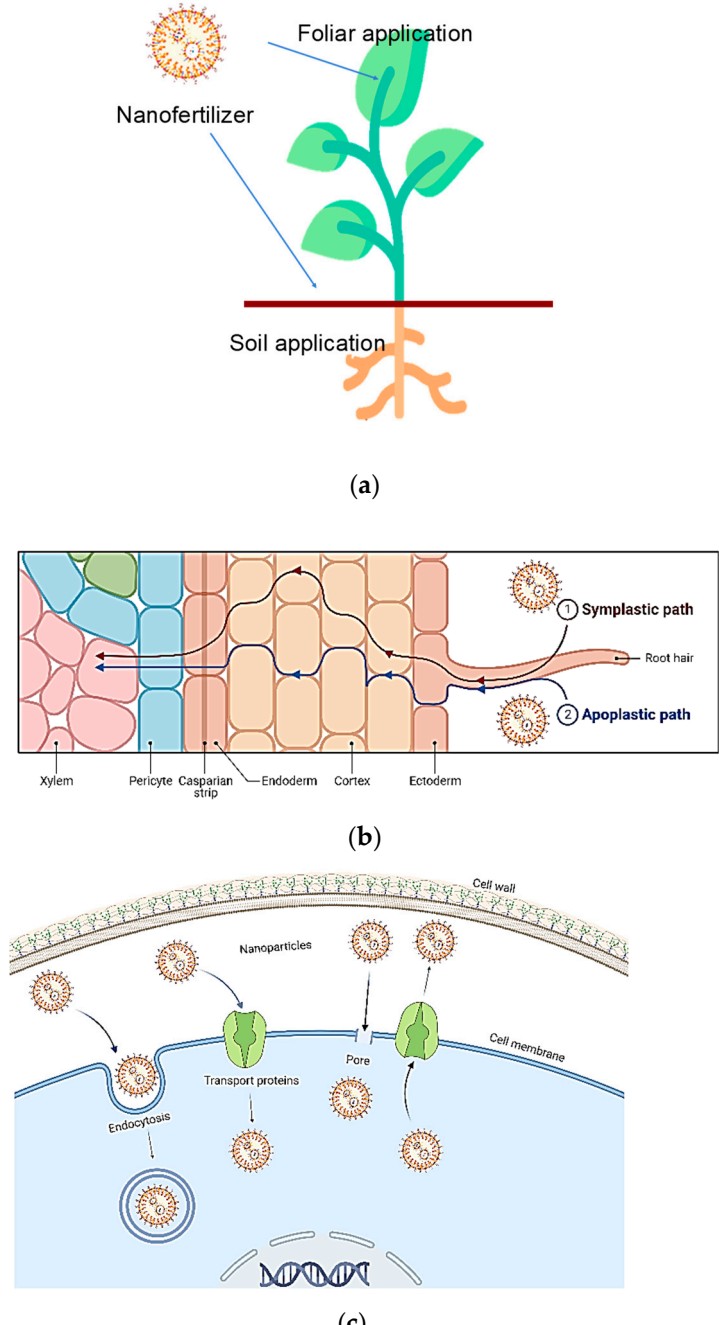

**Figure 5.** Route of nanofertilizer entry into plant and plants cells. (**a**) Nanofertilizer applied to leaves or soil. (**b**) Route of nanofertilizer entry in plants. (**c**) Nanoparticle entry in plant cells.

The utilization of nanofertilizer with a targeted and controlled release system is designed to enhance nutrient delivery efficiency, reduce dosage and nutrient loss, as well as minimize soil and water pollution, to meet the key characteristics and needs in the environmental, social, health, and economic realms (Figure 5). These techniques can decrease the need for irrigation and use artificial fertilizers and pesticides on crops, saving energy and resources [189]. In nanoformulations, additives (stabilizers) play a significant role as controlled release carriers and agents for protection, dispersion, delivery, or photocatalysis. Another strategy is to utilize emulsions (nanoemulsions, microemulsions, and nanodispersion) to boost the solubility of active substances.

## 4. Modes of Nanofertilizer Application

There are three primary methods of nanofertilizer application: foliar, seed nanopriming, and soil treatment. The foliar application involves spraying nanofertilizers directly onto the leaves of plants, allowing for rapid nutrient absorption through the leaf surface [7]. The method is particularly effective when nutrients are required quickly or in regions with low soil fertility. However, foliar application is sensitive to environmental factors such as temperature, humidity, and wind, affecting nutrient uptake efficiency [190]. Seed nanopriming entails coating or soaking seeds in a solution containing nanofertilizers before planting [191]. The method promotes rapid germination, stronger seedlings, and enhanced nutrient uptake throughout the plant's life. It is especially beneficial in areas with poor soil quality or where rapid plant establishment is necessary. However, the optimal concentration of nanofertilizers must be determined to avoid phytotoxicity [31]. Soil treatment involves incorporating nanofertilizers directly into the soil by broadcasting, banding, or localized placement [159]. The method ensures the slow and controlled release of nutrients, reducing nutrient loss through leaching or volatilization. Soil treatment is best suited for regions with high nutrient retention capacities and climates with consistent precipitation patterns. However, the application must be carefully managed to prevent nutrient imbalances or environmental pollution [8].

The appropriate method of nanofertilizer application is crucial for optimal plant growth, as it varies depending on the soil and climate type. The choice depends on soil quality, nutrient availability, and climate, which affect nutrient uptake and utilization. Understanding these factors and selecting the appropriate method can improve crop yield, reduce environmental impact, and create more sustainable agricultural practices.

The three methods of application are explained below in detail.

### 4.1. Foliar Spray

Foliar spray is an advanced method that directly applies liquid fertilizers to plants' leaves or foliage, enabling rapid absorption of nutrients through the leaf surface. The method employs nanofertilizer delivery to the leaf surface for targeted, optimal, rapid, and accurate transfer to the plant. The foliar application of NPs has emerged as a promising method for delivering essential elements such as nanofertilizers, fungicides, herbicides, and preservatives to plants. This approach leverages delayed release mechanisms to enhance the effectiveness of these substances. The absorption of foliar-applied NPs can occur through stomata, endocytosis, and direct absorption, although the process heavily depends on particle size. Leaf wax and cell walls can act as barriers, inhibiting the uptake of these particles. Once absorbed, the majority of NPs accumulate in vacuoles. However, various factors influence the absorption and transport of NPs, including plant characteristics, NP physical properties, and environmental conditions.

The foliar spray offers several advantages over traditional soil applications, including a faster response, improved nutrient utilization, and reduced leaching and run-off [192]. Multiple studies have demonstrated that foliar application of nanofertilizers can significantly improve nutrient uptake, promote plant growth, and increase crop yield. Foliar application of CeO and carbon-based NPs increased wheat yield by 36.6% [193] and bitter melon yield by 28% [194]. Another study reported that foliar application of copper NPs in tomato plants increased fruit yield by 80%, with a 30% reduction in the required copper concentration compared to conventional copper-based fungicides [195].

### 4.2. Seed Nanopriming

Seed priming is a pre-sowing treatment that induces physiological changes within seeds, allowing for faster germination and promoting plant growth and development by regulating metabolic and signaling cascades. The method involves soaking seeds in nanofertilizers, which has been shown to reduce fertilizer application by half while achieving excellent results [196]. Nanobiofertilizers act as stimulants, enhancing germination and

development by penetrating seed pores, dispersing within, and activating plant hormones that promote growth.

Applying nanofertilizer to seed priming increases seed germination by eliminating reactive oxygen species and regulating plant development hormones [172]. Seed priming also stimulates the expression of multiple genes during germination, particularly those related to plant resilience, resulting in enhanced resistance [7]. Conventional seed priming methods employ water, nutrients, or hormones to dissolve a seed coat. In contrast, advanced seed nano-priming techniques involve applying nanofertilizers directly to the seed surface, leaving a substantial fraction impeding pathogen penetration.

Nano-compound absorption at the cellular level reduces input and avoids molecular interactions, allowing for the production of highly resistant seeds with improved germination and seedling growth, especially under stress. Studies have shown that bean seed priming with chitosan NPs (0.1, 0.2, and 0.3%) for 3 h, followed by 100 mM NaCl treatment, enhanced seed germination and radicle length [197]. Under salt stress, proline, chlorophyll a, and antioxidant enzyme efficiencies of bean seedlings treated with 0.1% chitosan NPs increased significantly compared to untreated, salt-stressed seedlings [197]. Nanofertilizers mitigate plant stress by regulating internal hormone action in crops, strengthening antioxidants, and reducing reactive oxygen species (ROS) formation [172].

### 4.3. Soil Treatment

Nanofertilizers can be administered to the soil using conventional techniques such as broadcasting, side-dressing, or fertigation methods. Once in the soil, the NPs interact with plant roots through adsorption to the root surface or by penetrating root cells via endocytosis [198,199]. When applied to soil, nanofertilizers can interact with plants, soil particles, and microorganisms, which may alter their behavior and function. The controlled release of nutrients from the NPs ensures a steady supply of essential elements, which enhances plant growth and productivity [35]. This method of application, although considered reliable, suffers from uncertain long-term effects of NPs [69], higher costs [187], and regulator challenges [200].

## 5. Advantages of Nanofertilizers over Conventional Chemical Fertilizers

Nanofertilizers demonstrate various advantages compared to traditional fertilizers, including increased efficiency due to the direct delivery of essential nutrients to plants and decreased environmental impacts through reduced required fertilizer amounts. This technology has the potential to not only maximize crop yields but also decrease the environmental effects of fertilizers.

The advantages of nanofertilizers include their high nutrient concentration, slow release of nutrients, and improved plant uptake (Table 3 and Figure 6). Nanofertilizers can also enhance the physical and chemical properties of soils, and their use can help to reduce fertilizer use and the environmental impacts of agriculture. Nanofertilizers boast high nutrient concentrations, enabling lower application rates than their traditional counterparts. As a result, fertilizer costs can be reduced, and the associated environmental impacts from production and transportation are mitigated. Slow-release nanofertilizers can provide a steady supply of nutrients to plants over an extended period, improving plant growth and yield. Such nanofertilizers can also help reduce nutrients' leaching into the environment and the need for frequent reapplication of fertilizers. Improved uptake of nutrients by plants can lead to increased growth and yield and reduced nutrient losses to the environment. Nanofertilizers can help improve fertilizer efficiency, and their use can reduce the overall environmental impact.

**Table 3.** Advantages and disadvantages of nanofertilizers over conventional fertilizers.

| Properties | Nano Fertilizers | Conventional Fertilizers |
|---|---|---|
| Nutrient uptake efficiency | Increases fertilizer utilization efficiency and the ratio of plant nutrient uptake while saving fertilizers. | Less effective since its bulk composites are poorly absorbed by plants. |
| Control release modes | Encapsulation, in conjunction with a covering of polymer resin, waxes, and sulfur, permits precise control over the release of nutrients. | Excessive release results in toxicity and undermines ecological balance. |
| Solubility and dispersion of nutrients | Increases the solubility and dispersion of insoluble mineral components in soil, making them more bioavailable to plants. | Less available to plants due to lower solubility and larger particle size |
| Effective duration of release | Improves and prolongs the plant's nutrient acquisition rate | During delivery, nutrients required by plants are lost as insoluble salts. |
| Low rate of fertilizer needed | Reduces nutrient losses resulting from leaching, runoff, and drift. | High fertilizer levels are lost due to leaching, runoff, and drift. |

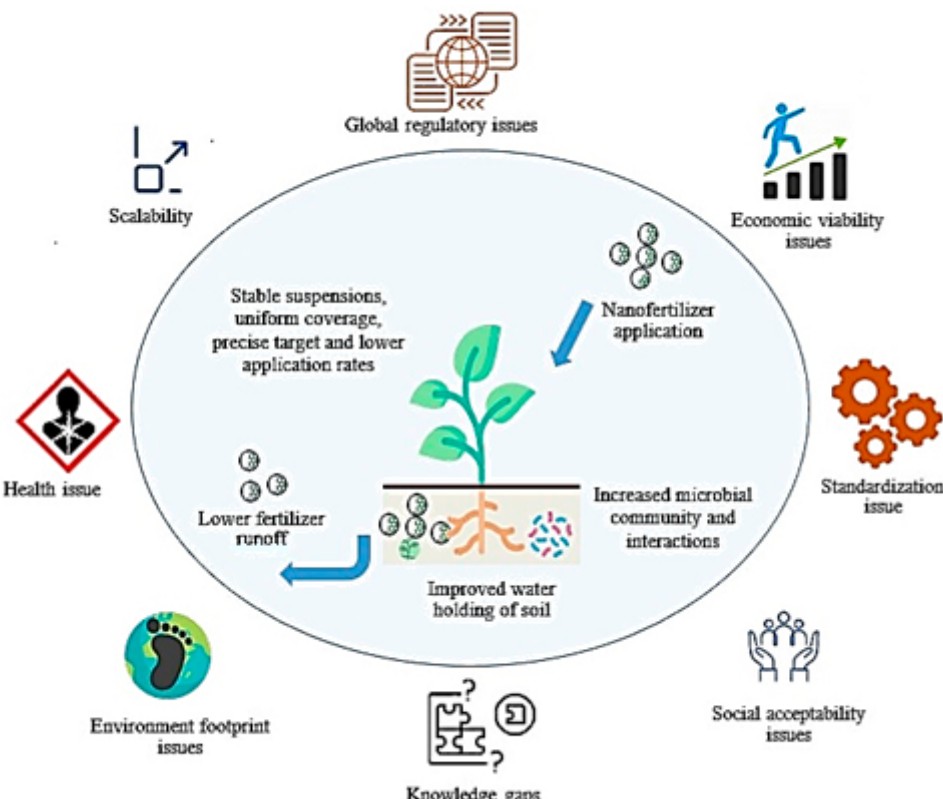

**Figure 6.** Advantages and issues of nanofertilizer application in the field.

Nanofertilizers provide several advantages over traditional fertilizers, some of which are listed below:

### 5.1. Greater Surface Area

The particle size of nanofertilizers is less than 100 nm, which increases their capacity to penetrate plants from applied surfaces such as soil or leaves, boosting plant nutrient uptake [201], which supports a broader range of sites and diverse metabolic functions in plant systems, leading to more photosynthate production. Increased surface area improves nutrient uptake and utilization efficiency while increasing the reactivity of nanofertilizers with other substances [202].

## 5.2. High Solubility

The solubility of a substance refers to the maximum amount that can be dissolved in a given solvent at a specific temperature and pressure. Nanofertilizers exhibit increased solubility due to their reduced particle size and larger surface area, which promotes dissolution in the soil solution [31]. Nanofertilizers can dissolve easily in many solvents, such as water, resulting in the increased solubility of insoluble nutrients in the soil and enhanced availability of the nutrients to the organisms in the environment [203]. The high solubility of nano-hydroxyapatite, a phosphorus-based nanofertilizer, has been demonstrated compared to conventional phosphorus fertilizers [204]. The solubility of nano-zinc oxide, a zinc-based nanofertilizer, is significantly higher than that of bulk zinc oxide [205].

## 5.3. Encapsulation of Fertilizers within NPs

Due to the encapsulation of fertilizers in NPs, nutrient absorption in crops increases [206]. The methods for encapsulating fertilizers include polymer-based encapsulation by forming a polymer matrix around the fertilizer through emulsion polymerization and interfacial polymerization [207]. Another method is inorganic shell formation by incorporating inorganic materials, such as silica or calcium carbonate, which can form a shell around the fertilizer through sol-gel synthesis or precipitation [8]. Yet another method is layer-by-layer assembly, which involves the sequential deposition of charged polymers, NPs, or other materials onto the fertilizer surface, forming a multilayered shell [208].

For example, zeolite-based nanofertilizers are used for nanoencapsulation of fertilizers. This approach uses zeolites as nanoscale carriers to encapsulate nutrients within their porous structure. Zeolite-based nanofertilizers boost nutrient availability throughout crop development, reduce nutrient loss via denitrification, volatilization, and leaching, and fix nutrients in the soil, particularly $NO_3$ and $NH_4^+$ [209].

## 5.4. Easy Penetration and Controlled Release of Fertilizers

Nanofertilizers significantly aid the penetration of fertilizers into plant tissues, enhancing nutrient uptake and utilization by plants [8]. The improved penetration is primarily attributed to the nano-sized particles that comprise nanofertilizers, which possess a high surface area–to–volume ratio [31]. These smaller particles can quickly diffuse through plant cell walls and membranes [7]. Furthermore, the functionalization of these NPs with specific targeting molecules enables them to be selectively taken up by plant tissues, ensuring a precise delivery of essential nutrients [210]. The improved penetration of nanofertilizers reduces the amount of fertilizer required and minimizes nutrient leaching and runoff, thereby mitigating environmental pollution [9].

Consequently, using nanofertilizers for efficient penetration in plant tissues contributes to better crop yields, improved nutritional quality, and overall agricultural sustainability [211]. Nanofertilizers serve a crucial function in the improved availability of nutrients to the plant and, consequently, in the healthy growth of seedlings due to their high penetration rate. For example, nano-ZnO exhibits more peanut seed germination and root development than zinc sulfate in bulk [212].

## 5.5. High Nutrient Absorption Efficiency

Nanofertilizers increase fertilizer efficacy and the ratio of soil nutrient absorption in crop yield, hence decreasing fertilizer use. Additionally, nanofertilizers prevent fertilizer leaching loss [213]. In a study, applying zinc oxide NPs significantly increased wheat plants' zinc uptake, root length, and shoot dry weight compared to traditional zinc sulfate fertilization [214]. In another study, using iron oxide NPs increased iron uptake and chlorophyll content in wheat plants, resulting in higher yields than conventional iron fertilizers [215]. A test of nitrogen release in both acidic and neutral soils revealed that the nano-urea-HAP mixture releases urea at 44% lower rates at pH 7 in 60 days study compared to regular urea [216].

*5.6. Effective Duration of Nutrient Release*

The effective duration of nutrient release in nanofertilizers has significant implications for agricultural productivity and sustainability. It ensures that plants receive the necessary nutrients throughout the growth cycle. Moreover, the controlled release of nutrients minimizes nutrient losses, reducing the need for repeated fertilizer applications and mitigating environmental pollution [69].

Bulk fertilizers are only effective for a short time after application. However, nanofertilizers may enhance the nutrition release duration [213]. Several studies have investigated the effective duration of nutrient release in nanofertilizers. For example, a study reported that chitosan-based nanofertilizers released nutrients for up to 45 days [9]. Another study demonstrated that polymer-coated nanofertilizers exhibited a nutrient release duration of up to 60 days [163]. Nanoparticle size [211], polymer coating [159], and environmental factors such as soil type, pH, temperature, and moisture [7] affect the effective duration of nutrient release in nature.

*5.7. Improved Microbial Activity*

The interaction between NPs and microorganisms, the shelf life of biofertilizers, and the dissemination of nanofertilizers are among the most crucial variables in plant growth. Nanobiofertilizers combine NPs and living microorganisms designed to improve plant growth and yield by enhancing nutrient uptake and soil fertility. It improves the structure and function of soil and the morphological, physiological, biochemical, and yield attributes of plants. The formation and application of nanofertilizer is a practical step toward smart fertilizer that enhances growth and augments the yield of crops.

The interaction between gold NPs and plant growth-promoting rhizobacteria is advantageous to plants [177,217]. In another study chitosan based NPs were used to deliver nitrogen-fixing bacteria to plants. Coating nitrogen-fixing bacteria with chitosan NPs facilitates efficient penetration into plant roots [218]. This enhanced infiltration enhances nitrogen availability to the plants, resulting in enhanced growth and increased yield [219]. As a consequence, this approach mitigates the adverse impacts on microbial biological processes and minimizes disturbances to cell membrane structure and functions.

However, the biofertilizer shelf life is a limiting factor in these formulations, and nanomaterials can be utilized to extend it. Nanoformulations can enhance biofertilizer resilience to desiccation, heat, and UV damage. Research suggests that NPs can improve plant health during abiotic and biotic stress conditions through rhizobacterial metabolite changes [220].

*5.8. Improved Soil Activity*

Soil activity is a measure of the biological activity in the soil, including decomposition and nutrient cycling. These processes are essential for maintaining healthy soil and are driven by soil microorganisms, such as bacteria and fungi. Nanofertilizers increase soil activity by providing efficient delivery of nutrients to the root zone. Nanofertilizers increase the activity of soil microorganisms, leading to improved decomposition of organic material and increased nutrient cycling [221,222]. Nanofertilizers increase the nitrogen mineralization rate in soil, a crucial process for maintaining soil fertility [223,224]. They improve the activity of soil bacteria and fungi, essential for maintaining the soil ecosystem [225].

*5.9. Improved Soil Water-Holding Capacity*

Nanofertilizers improve soil structure and water-holding capacity, increase organic matter, and create favorable conditions for beneficial microorganisms. They also contain humic acid and clay, which bind soil particles and reduce water loss through runoff and evaporation, resulting in better crop yields and improved soil health [226]. Moreover, some nanofertilizers, particularly those based on clay and carbon NPs, can bind soil particles together, forming larger aggregates [184]. The improved soil aggregation leads to better soil structure, enhancing water-holding capacity and reducing soil erosion [69].

### 5.10. Ecofriendly Nature

Besides being more efficient, nanofertilizers are much safer for the environment [31, 150]. Traditional fertilizers release large amounts of nitrogen and phosphorus into the soil, which can cause eutrophication and algal blooms in nearby water bodies [185]. Nanofertilizers, on the other hand, are designed to release their nutrients slowly over time, allowing for more nuanced control of the nutrient balance in the soil and reducing the risk of environmental damage from nutrient runoff [8]. Since they are more efficient than traditional fertilizers, they require less input, meaning farmers can save money on fertilizer costs. Furthermore, the slow release of nutrients provided by nanofertilizers reduces the need for frequent applications and labor costs [157]. As nanotechnology continues to improve, these fertilizers will become even more efficient and cost-effective, allowing farmers to produce more with less.

### 5.11. Low Production Cost

Nanofertilizers could attain lower production costs due to enhanced nutrient use efficiency, controlled release, and targeted delivery, reducing the wastage of fertilizers in the field. Nanofertilizers are typically much cheaper than conventional fertilizers as they are less labor-intensive, need lower fertilizer per application, and have higher absorption rates than traditional fertilizers. Furthermore, nanofertilizers can remain in the soil for extended periods, resulting in fewer applications and lower costs [227].

### 5.12. Fulfills the Goal of Precision Farming

Precision farming is an agricultural production management system that uses information technology to measure and manage crop production inputs such as fertilizers, pesticides, and water. Recent advances in nanotechnology have allowed nanofertilizers to develop, which may be used to improve precision farming practices [69]. The use of nanofertilizers for precision farming offers several potential benefits. Firstly, nanofertilizers can be directly applied to the plant, enabling a more precise application than traditional fertilizers. This results in reduced costs, lower fertilizer input, and a lesser runoff volume that may enter the environment. In addition, nanofertilizers can deliver specific nutrients to the plant, allowing for more precise crop nutrition management. Finally, nanofertilizers can also provide other substances in the plant, such as micronutrients and growth regulators.

### 5.13. Improves Plant Stress Tolerance

Plants are affected by numerous environmental stresses throughout their life cycle. As a result, they modify genetic, biochemical, and physiological pathways to strengthen their defense against environmental stresses in various phases. Plants respond to such types of abiotic stress by altering gene expression at the molecular level. Multiple studies show that NPs' impact on plant growth and development is dose-dependent. In plants, the signaling system excites the defense machinery, which activates molecular mechanisms to respond to diverse stress circumstances. Nanofertilizers have been found to improve plant stress tolerance in several ways [228]. The nanoscale particles of the fertilizers can penetrate the cell walls of plants, allowing nutrients to reach the root system and other parts of the plant more quickly. In addition, nanofertilizers may also help plants better tolerate environmental stressors, such as drought or extreme temperatures. For example, nanofertilizers can provide essential nutrients that help plants remain healthy despite water shortages or heat waves. Moreover, nanofertilizers can also offer beneficial protective compounds, such as antioxidants, that can help plants cope with environmental stressors.

### 5.14. Stimulates Plant Growth

NPs increase the availability of essential nutrients such as nitrogen, phosphorus, and potassium, which are necessary for the growth and development of plants. NPs also assist in the uptake of these essential nutrients, allowing plants to utilize them more efficiently [159]. Nanofertilizers also enhance soil structure and texture and improve water retention. Nanofer-

tilizers create a protective layer on the soil surface, which can help to reduce the loss of essential nutrients. In addition, nanofertilizers can help maintain the soil pH balance, positively affecting plant growth [163]. Nanofertilizers can deliver nutrients to the plant root zone more efficiently and penetrate the soil surface more effectively, allowing for better nutrient absorption. Furthermore, nanofertilizers reduce the amount of fertilizer runoff, which can harm the environment [228]. All these factors help in stimulating plant growth.

## 6. Limitations and Potential Risks Related to the Application of Nanofertilizers

Despite the positive findings, various limits and detrimental consequences have been documented when utilizing nanofertilizers [169]. Most nanofertilizer studies have only been conducted in the lab or on a small scale [11]. Foliar application of nanofertilizers has the drawback of needing a large usable leaf area and the risk of scorching or burning if the spray concentration is too high [229]. Due to the influence of weather on their efficacy, they must be applied at the precise right time. Optimization of foliar applications of nanofertilizers, standardization of the nanoformulations, and lack of size homogeneity of the nanoparticles are all issues that need further investigation [169].

The impact of nutrition in supplying such molecules in pastures through a transformation within the plant and the effect on the environment is poorly understood. It is unknown whether all nanofertilizers are converted to ionic forms in the plant and then incorporated into proteins and other metabolites or whether some survive intact and reach consumers through the food chain; this remains an open question [169].

To gain a comprehensive understanding of nanomaterials' advantages, it is imperative to acquire additional and improved data on materials characterization, conduct detailed comparisons with non-nano formulations, and carry out field investigations [11].

Nanotechnology is a rapidly growing field with numerous applications in agriculture, including developing nanofertilizers. These nanofertilizers can enhance nutrient availability, reduce nutrient losses, and promote plant growth [8]. However, the potential risks of applying nanofertilizers have not been thoroughly explored. This section discusses these risks and emphasizes the need for further research to address these concerns.

### 6.1. Human Health Risks

One of the primary concerns associated with nanofertilizers is their potential impact on human health. Due to small size, organisms can easily absorb NPs, leading to potential toxicity [230]. Studies have shown that the ingestion of nanofertilizers can cause damage to the gastrointestinal tract, liver, and kidneys in experimental animals [231]. Furthermore, NPs can cross biological barriers, such as the blood-brain barrier, potentially causing neurological damage [230]. Further research is needed to understand the long-term effects of nanofertilizer exposure on human health.

### 6.2. Environmental Risks

The release of NPs into the environment may lead to contamination of soil, water, and air [9]. NPs can accumulate in the soil, potentially disrupting soil ecosystems and decreasing soil fertility [232]. Moreover, leaching NPs from the soil into aquatic ecosystems could adversely affect aquatic organisms, leading to bioaccumulation and biomagnification within the food chain [233]. The potential risks associated with nanoparticle release into the environment require further investigation to minimize negative consequences.

### 6.3. Ecological Risks

Another major concern regarding nanofertilizers is their potential impact on non-target organisms. Studies have demonstrated that exposure to NPs can cause adverse effects in various organisms, including insects, fish, and birds [234]. Nanofertilizers can interfere with organisms' reproduction, growth, and development, potentially leading to population declines [235]. The impact of nanofertilizers on beneficial microorganisms, such

as nitrogen-fixing bacteria and mycorrhizal fungi, is not yet fully understood [236]. Further research is necessary to evaluate the potential ecological risks of nanofertilizer application.

## 7. Current Status and Future Outlook

Using nanoparticles in fertilizers promises efficient nutrient delivery to plants, enhancing crop productivity and reducing the environmental impact of conventional fertilizers. Researchers are extensively exploring various nanomaterials, such as nanozeolites, nanochitosan, and metal oxide nanoparticles, for their potential to improve nutrient absorption and retention in soils. Significant breakthroughs have been made in developing controlled-release nanofertilizers that gradually supply nutrients, reducing the frequency of application and potential nutrient losses.

Prioritizing important research and development areas is necessary to create and use nanofertilizers in agriculture. The prospects of nanofertilizers in agriculture are promising due to their numerous advantages. However, future research in nanofertilizers should focus on developing eco-friendly and cost-effective synthesis methods for various nanomaterials, optimizing their physicochemical properties, and minimizing potential risks associated with their use. To create long-lasting, biodegradable, and sustainable products for the future, we need green technology innovation (GTI) and Internet of Things (IoT) technologies. All innovations significantly improving natural resource utilization and reducing environmental harm, impact, and worsening are considered part of GTI [237]. Efforts are to be made to design formulations that enable the controlled release and targeted delivery of nutrients using "smart" nanofertilizers that can respond to specific environmental cues, such as pH or temperature, to release nutrients when and where they are needed [238]. This approach can help reduce nutrient losses and enhance plant nutrient use efficiency, promoting sustainable agricultural practices.

Integrating nanosensors into nanofertilizers could enable real-time soil nutrient level monitoring, allowing for precise application and reduced nutrient waste. This technology can help farmers apply the correct amount of fertilizer at the right time, thus improving crop yields and reducing environmental pollution [239].

In addition, the research should focus on understanding the potential risks of nanofertilizers to the environment and human health, as well as developing clear regulatory frameworks and standardized guidelines to ensure their safe and effective use. There is a need to investigate the potential toxicity of nanofertilizers on soil organisms, crops, and humans to ensure their safety in the long term. Lastly, education and outreach initiatives targeting farmers and other stakeholders are critical in fostering the acceptance and use of nanofertilizers in agriculture, which can be achieved through disseminating information about the benefits of nanofertilizers and their safety and safety and effective use [240,241].

Precision agriculture, involving drones fitted with cameras to acquire multispectral pictures detecting nutrient concentration in the field, can help farmers avoid wasting precious resources by accidentally applying too much to their crops or too often. To mitigate risks and maximize benefits from opportunities and synergies that nanofertilizer use may present should be studied for the long-term viability of agricultural lands. However, developing and using these molecules to increase food production with increased nutrient efficiency requires weighing the production economics and environmental costs against the potential ecological consequences and yield benefits. To this end, a life cycle assessment of nanoparticles could provide a holistic evaluation of their application by considering their yield productivity, environmental implications, and effect on food chains. Another unfinished challenge before the widespread use of nanofertilizers is validating their benefits and drawbacks in realistic field settings to address stakeholders' concerns.

The current review retrieved 310 articles from PubMed, searching the keyword "nanofertilizers" (Figure 7a). The number of articles published is a strong indicator of the health of a field and its ability to attract researchers. From 2019 onwards, there was a sharp increase in articles discussing nanofertilizers. Figure 7 illustrates the graphical representation of the network visualization map for the co-occurrence of the most cited

keywords along with "nanofertilizer" from 2013 to 2023. According to the illustration, nanoparticles, agriculture, and nanofertilizer are among the most investigated fields, with more research concentrating on creating novel nanofertilizers.

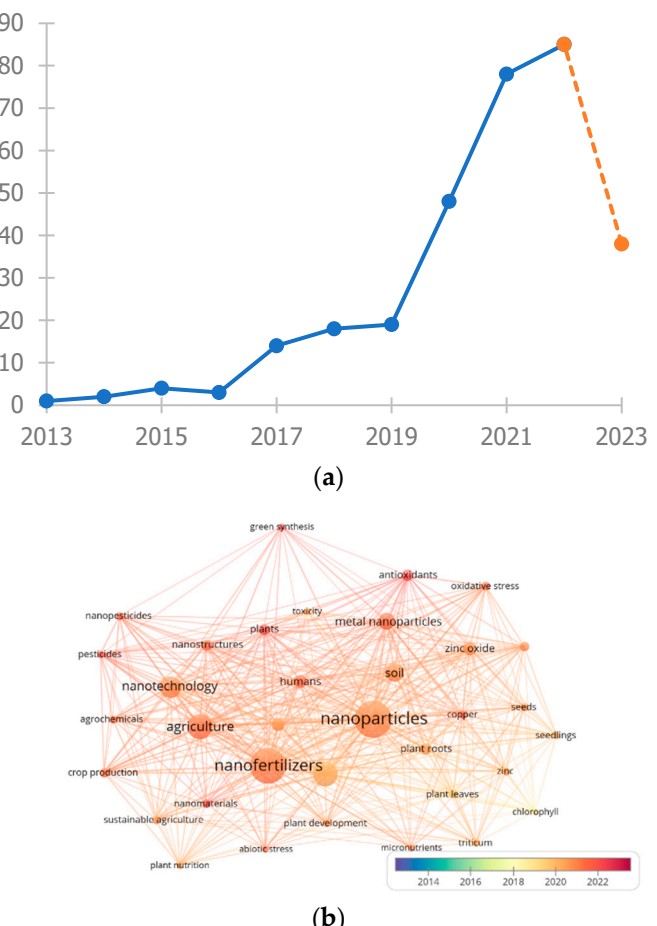

(**a**)

(**b**)

**Figure 7.** (**a**) Number of agriculture-based scientific articles published on nanofertilizers over the past decade (the red dotted line represents incomplete data for 2023). (**b**) Ten-year overlay visualization of associated 35 keywords having at least ten co-occurrences. The size of the nodes represents the keyword frequency in the article, while the connecting lines reflect the co-citation network.

Despite these advances, safety concerns associated with using nanomaterials on human health and the environment remain. The studies should also focus on the environmental fate, toxicity, and biodegradability of nanofertilizers. The potential for nanofertilizers is vast, but robust regulatory frameworks and in-depth risk assessments are required to ensure sustainable and safe usage.

## 8. Conclusions

Nanofertilizers have emerged as a promising tool for sustainable agriculture and global food security, offering numerous advantages over conventional fertilizers. These advantages include controlled release, targeted delivery, plant growth stimulation, and reduced water and nutrient loss. With the global population expected to exceed 9 billion by 2050, the development of nanofertilizers could become essential in meeting the growing demand for food while minimizing environmental impacts.

Various methods are being explored to create nanofertilizers, such as nanoencapsulation and NPs, for direct nutrient delivery to plant cells. Despite the potential benefits, much research is needed to optimize nanomaterial composition and release rates and develop scalable and cost-effective production methods. Addressing these challenges will make nanofertilizers accessible and affordable for farmers worldwide, especially in

developing countries where food security and sustainable agriculture are pressing concerns. While the potential benefits of nanofertilizers are significant, addressing the potential risks associated with their application, such as human health, environmental, and ecological risks, is essential. Developing regulations and guidelines for their safe and responsible use will ensure agricultural sustainability and protect human health and the environment. Interdisciplinary collaboration between researchers, agricultural experts, and policymakers is vital for developing and implementing nanofertilizers. This collaboration will facilitate knowledge exchange, improve the understanding of the implications of nanofertilizers in the agricultural ecosystem, and help develop best practices for their usage. Furthermore, educating farmers about the advantages, application methods, and potential risks of using nanofertilizers will enable them to make informed decisions about adopting these innovative products in their farming practices.

Continuous monitoring and assessment of the impacts of nanofertilizers on crop yields, soil health, and the environment is necessary to make data-driven decisions and adjustments as technology advances, which will ensure that the benefits of nanofertilizers are maximized while minimizing negative consequences. Future research should focus on refining the delivery mechanisms, exploring new materials and strategies for controlled release, and assessing long-term effects on human health and the environment. Ultimately, the successful integration of nanofertilizers into modern agricultural practices can revolutionize how we approach sustainable agriculture, ensuring global food security while minimizing our ecological footprint.

In conclusion, developing nanofertilizers presents an exciting opportunity to revolutionize the agricultural sector and promote sustainable practices. By focusing on ongoing research, addressing potential risks, fostering interdisciplinary collaboration, educating farmers, and ensuring affordability and accessibility, the potential of nanofertilizers to contribute to global food security and agricultural sustainability significantly can be realized.

**Author Contributions:** A.Y. drafted the manuscript. K.Y. redrafted the manuscript, while K.A.A.-E. was responsible for the revision of the manuscript. All authors have read and agreed to the published version of the manuscript.

**Conflicts of Interest:** The authors declare no conflict of interest.

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
