# Peer review of "Nanofertilizers: Types, Delivery and Advantages in Agricultural Sustainability"

_agrochemicals, doi:10.3390/agrochemicals2020019_

Round 1

Reviewer 1 Report

The review focuses on the current trends in the use of nano-fertilizers. The manuscript is clearly written and brings a vast information related to this topic.

To improve the quality of this paper, it is recommended to make the following revisions:

- Please refer to the figures and table in the text.

- Please amend the paper with specific data (values). It is advisable to convert the sections 4 and 5 into the table with specific values.

- The potential risks related to the application of nano-fertilizers, should be discussed.

- Please pay attention to the comments inserted directly in the file.

Kind regards

The English used correct and readable (the reviewer is a non-native English speaker).

Author Response

Comments and Suggestions for Authors

The review focuses on the current trends in the use of nano-fertilizers. The manuscript is clearly written and brings a vast information related to this topic.

To improve the quality of this paper, it is recommended to make the following revisions:

  1. Please refer to the figures and table in the text.

Figures and tables modified

  1. Please amend the paper with specific data (values). It is advisable to convert the sections 4 and 5 into the table with specific values.

Sections 4 and 5 have been merged

  1. The potential risks related to the application of nano-fertilizers, should be discussed.

Potential risks added in section 5.

  1. - Please pay attention to the comments inserted directly in the file.

No comment was found inserted in file

Reviewer 2 Report

This topic is very important but there already lost of review on this subject.

Therefore there is overlapping with paper: Babu, S.; Singh, R.; Yadav, D.; Rathore, S. S.; Raj, R.; Avasthe, R.; Yadav, S. K.; Das, A.; Yadav, V.; Yadav, B.; Shekhawat, K.; Upadhyay, P. K.; Yadav, D. K.; Singh, V. K. Nanofertilizers for Agricultural and Environmental Sustainability. Chemosphere 2022, 292, 133451. https://doi.org/https://doi.org/10.1016/j.chemosphere.2021.133451, which is not even cited.

Therefore, I suggest rejection and resubmission after reconstruction of the manuscript.

New references should be added, for example, controlled release of urea even with gypsum has already been published (Scale-up of agrochemical urea-gypsum cocrystal synthesis using thermally controlled mechanochemistry // ACS Sustainable Chemistry & Engineering, 10 (2022), 20; 6743-6754 doi:10.1021/acssuschemeng.2c00914)

Add citations and avoid overlapping with similar reviews as much as: Nongbet, A.; Mishra, A.K.; Mohanta, Y.K.; Mahanta, S.; Ray, M.K.; Khan, M.; Baek, K.-H.; Chakrabartty, I. Nanofertilizers: A Smart and Sustainable Attribute to Modern Agriculture. Plants 202211, 2587. https://doi.org/10.3390/plants11192587

This review have to be improved.

Indeed, in some chapters the advantages and disadvantages are described, while in others they are not. The logic of the listing of facts in Part 2 is not entirely clear. For example, the sentences in Sulfur based refer to references 84, 85, 86, but the short comment why and how should be added.

Part 3-5 you can comment and/or add studies and results on possible disadvantages

References: Add the Doi number wherever possible

The English language has to be improved

The English language has to be improved for clarity.

Sentences could be simplified and clarified by English proofreading.

Author Response

Comments and Suggestions for Authors

This topic is very important but there already lost of review on this subject.

  1. Therefore there is overlapping with paper: Babu, S.; Singh, R.; Yadav, D.; Rathore, S. S.; Raj, R.; Avasthe, R.; Yadav, S. K.; Das, A.; Yadav, V.; Yadav, B.; Shekhawat, K.; Upadhyay, P. K.; Yadav, D. K.; Singh, V. K. Nanofertilizers for Agricultural and Environmental Sustainability. Chemosphere 2022, 292, 133451. https://doi.org/https://doi.org/10.1016/j.chemosphere.2021.133451, which is not even cited. Therefore, I suggest rejection and resubmission after reconstruction of the manuscript. I would like to bring to your attention the differences between the published article and my article, and explain why my article is an important addition to the literature and deserves to be published.

Answer: The said article was not referred will preparing the manuscript due to the unavailibilty of full text. After your comment I requested the paper from the author and cited it. As far as similarity with the said article is concerned, both articles focus on nanofertilizers and their role in agricultural sustainability, approach the topic from distinct angles and provide unique insights. The published article is a comprehensive review that primarily categorizes nanofertilizers based on the nutrients they deliver (macronutrients and micronutrients) and also discusses nanobiofertilizers. This article thoroughly examines the effects of nanofertilizers on crop growth, productivity, quality, and nutrient content, as well as their toxicity, absorption, translocation, impact on plant stress tolerance, soil health, and environmental fate.

In contrast, my article, takes a different approach to classification, dividing nanofertilizers into action-based, nutrient-based, and consistency-based types. The article delves into materials and strategies for controlled and targeted delivery of nanoparticles, and outlines the qualities of an effective nanofertilizer. Furthermore, it contrasts the advantages of nanofertilizers over conventional fertilizers and addresses potential risks related to their application, such as human health, environmental, and ecological risks.

The unique classification systems and emphasis on controlled delivery strategies, qualities of effective nanofertilizers, and risks associated with their application in my article make it a valuable addition to the literature. This article complements the previous article by providing a fresh perspective on nanofertilizers, and it can serve as a guide for researchers and agricultural practitioners who want to develop, implement, and optimize nanofertilizers in a sustainable manner.

Please click the following link for the said article and see how different both articles are.

tinyurl.com/d9ps8mdh

  1. New references should be added, for example, controlled release of urea even with gypsum has already been published (Scale-up of agrochemical urea-gypsum cocrystal synthesis using thermally controlled mechanochemistry // ACS Sustainable Chemistry & Engineering, 10 (2022), 20; 6743-6754 doi:10.1021/acssuschemeng.2c00914)

Answer: The manuscript is vastly revised and many new reference added.

  1. Add citations and avoid overlapping with similar reviews as much as: Nongbet, A.; Mishra, A.K.; Mohanta, Y.K.; Mahanta, S.; Ray, M.K.; Khan, M.; Baek, K.-H.; Chakrabartty, I. Nanofertilizers: A Smart and Sustainable Attribute to Modern Agriculture. Plants 2022, 11, 2587. https://doi.org/10.3390/plants11192587

Answer: More citations added and overlapping avoided.

  1. This review have to be improved.

Answer: The article has been improved in language, latest citations added and also logic in conclusion section added.

  1. Indeed, in some chapters the advantages and disadvantages are described, while in others they are not. The logic of the listing of facts in Part 2 is not entirely clear.

Answer: Under “nanofertilizer types” in section 2 inclusion of facts is necessary as the unique nanofertilizer types are not discussed elsewhere. If facts are removed from this section I will lose point of comparison of quality and advantages of nanofertilizers.

For example, the sentences in Sulfur based refer to references 84, 85, 86, but the short comment why and how should be added.

Answer: The extra reference, which was not adding value to the manuscript, has been removed.

  1. Part 3-5 you can comment and/or add studies and results on possible disadvantages

Answer: Potential risk have been added in point 5

  1. References: Add the Doi number wherever possible

Answer: DOI added in more references

  1. The English language has to be improved

Answer: English grammar improved with editing software

Reviewer 3 Report

In this manuscript, authors reported a review paper that deals with nanofertilizers: types, delivery, and advantages in agricultural sustainability. They survey the progress achieved in this field by doing a classification of nanofertilizers. The manuscript presents some interesting knowledge in the field of nanofertilizers for agricultural sustainability. I reviewed the manuscript in a critical manner and some of the comments are given below:

General comments

The manuscript might be a contribution of interest for “Agrochemicals” and in principle within its specific scope, but it is not suitable for publication in this form. The manuscript lacks clear statements and critics from the authors and well stated outlook. The quality of writing is good with some grammar and spelling errors. The English language usage should be checked by a fluent English speaker and/or a professional language editing service.

Moreover, the Review mostly summarizes the works in the literature rather than provide some assessments of the reported results. A good review should distill key information rather than simply compile it. Finally, you don’t have an outlook section, a good review should be foresightful and forward looking. Please incorporate a detailed outlook of the field. I look forward to seeing your revised version.

Recommendation: Reconsider after minor revisions noted.

Specific comments

1.     Introduction needs to be improved.

2.     As I mentioned, the review only copy and summarize. It is difficult to find the authors’ deep thoughts and bullet critics about this field. In other words, there are so few comments about the published cited papers. For a better review, the negative and positive comments and suggestions are more important, which can point out the new directions from the previous publications.

3.     The potential citation points are so few. The authors should give out more potential citation points for readers which could easily be done by commenting on the negative and positive sides of the mentioned nanofertilizers.

4.     Authors are also not critical. A recommendation for the readers (from a more general standpoint) what would be the best combination (class or type of nanofertilizers) of choice for a given agricultural system would be good to have at the end of the work. What I am searching for is that the options presented are linked to the general problems.

5.     As I mentioned before, a review should be foresightful and forward looking therefore I suggest incorporating an outlook section where you describe the possible scenarios for future development of nanofertilizers for agricultural applications.

N/A

Author Response

Comments and Suggestions for Authors

In this manuscript, authors reported a review paper that deals with nanofertilizers: types, delivery, and advantages in agricultural sustainability. They survey the progress achieved in this field by doing a classification of nanofertilizers. The manuscript presents some interesting knowledge in the field of nanofertilizers for agricultural sustainability. I reviewed the manuscript in a critical manner and some of the comments are given below:

General comments

The manuscript might be a contribution of interest for “Agrochemicals” and in principle within its specific scope, but it is not suitable for publication in this form.

  1. The manuscript lacks clear statements and critics from the authors and well stated outlook.
  2. The quality of writing is good with some grammar and spelling errors. The English language usage should be checked by a fluent English speaker and/or a professional language editing service.
  3. Moreover, the Review mostly summarizes the works in the literature rather than provide some assessments of the reported results. A good review should distill key information rather than simply compile it.

Answer:

  1. Clarity imparted to introduction and conclusion section
  2. Grammar errors removed with language editing software
  3. The review covers a broader subject area. The assessment of result have been tried in summary. Also comparative tables have been added wherever possible.

Finally, you don’t have an outlook section, a good review should be foresightful and forward looking. Please incorporate a detailed outlook of the field. I look forward to seeing your revised version.

Answer: Outlook added

Recommendation: Reconsider after minor revisions noted.

Specific comments

  1. Introduction needs to be improved.

Answer: Introduction has been rewritten

  1. As I mentioned, the review only copy and summarize. It is difficult to find the authors’ deep thoughts and bullet critics about this field. In other words, there are so few comments about the published cited papers. For a better review, the negative and positive comments and suggestions are more important, which can point out the new directions from the previous publications.

Answer: New section of future outlook mentions about the vision of author

  1. The potential citation points are so few. The authors should give out more potential citation points for readers which could easily be done by commenting on the negative and positive sides of the mentioned nanofertilizers.

Answer: Negative and positive sides of nanofertilizers are now available in advantages and disadvantages table.

  1. Authors are also not critical. A recommendation for the readers (from a more general standpoint) what would be the best combination (class or type of nanofertilizers) of choice for a given agricultural system would be good to have at the end of the work. What I am searching for is that the options presented are linked to the general problems.

Answer: Entire manuscript critically revised. The possible best combinations added.

  1. As I mentioned before, a review should be foresightful and forward looking therefore I suggest incorporating an outlook section where you describe the possible scenarios for future development of nanofertilizers for agricultural applications.

Answer: Future outlook section has been added

Round 2

Reviewer 1 Report

The manuscript has been considerably improved.

Some minor revisions are suggested:

- L327. Table 1. Please provide the references in the table. Also Table 1 should be mentioned in the text.

- L964. Please check the table number.

-The manuscript contains three tables. Please check the numeration.

- L358-366. Please revise the numeration of sub-sections.

-The sub-section 5.1.1 is not found in the manuscript.

-L1498. Authors 4, 5, and 6 are mentioned, while only three co-authors are indicated at the beginning of the paper. Please check this section.

Kind regards

Author Response

The manuscript has been considerably improved.

  1. Some minor revisions are suggested:
    • Table 1. Please provide the references in the table. Also Table 1 should be mentioned in the text.
    • Please check the table number.
  2. -The manuscript contains three tables. Please check the numeration.
    • L358-366. Please revise the numeration of sub-sections.
  3. -The sub-section 5.1.1 is not found in the manuscript.
  4. -L1498. Authors 4, 5, and 6 are mentioned, while only three co-authors are indicated at the beginning of the paper. Please check this section.

Answers:

1. References added in Table 1. Table 1 mentioned in text. L964. Table number changed as per the serial number.

2.Numeration of tables changed. L358*366 numeration of sub-sections corrected.

3. Sub-section 5.1.1 is now 5.1

4. Three author details updated. 

Reviewer 2 Report

The authors have extensively revised and updated the review.

They answered the questions of the reviewers.

Therefore, I propose acceptance after minor correction:

L320 something is missing in the sentence, reference maybe, please correct 

L964 has to be deleted "Table 2. describes the advantages and disadvantages of each nanofertilizer type."

English is imporved.

Author Response

  1. L320 something is missing in the sentence, reference maybe, please correct 
  2. L964 has to be deleted "Table 2. describes the advantages and disadvantages of each nanofertilizer type."

Answers:

  1. Missing text added.
  2. The specified line text removed.